# A New Graph-Based Deep Learning Model to Predict Flooding with Validation on a Case Study on the Humber River

Victor Oliveira Santos [1,*], Paulo Alexandre Costa Rocha [1,2], John Scott [3], Jesse Van Griensven Thé [1,3] and Bahram Gharabaghi [1,*]

1    School of Engineering, University of Guelph, 50 Stone Rd. E, Guelph, ON N1G 2W1, Canada; pcostaro@uoguelph.ca (P.A.C.R.); jesse.the@weblakes.com (J.V.G.T.)
2    Mechanical Engineering Department, Technology Center, Federal University of Ceará, Fortaleza 60020-181, CE, Brazil
3    Lakes Environmental, 170 Columbia St. W, Waterloo, ON N2L 3L3, Canada; 18jbs2@queensu.ca
*    Correspondence: volive04@uoguelph.ca (V.O.S.); bgharaba@uoguelph.ca (B.G.)

**Abstract:** Floods are one of the most lethal natural disasters. It is crucial to forecast the timing and evolution of these events and create an advanced warning system to allow for the proper implementation of preventive measures. This work introduced a new graph-based forecasting model, namely, graph neural network sample and aggregate (GNN-SAGE), to estimate river flooding. It then validated the proposed model in the Humber River watershed in Ontario, Canada. Using past precipitation and stage data from reference and neighboring stations, the proposed GNN-SAGE model could estimate the river stage for flooding events up to 24 h ahead, improving its forecasting performance by an average of 18% compared with the persistence model and 9% compared with the graph-based model residual gated graph convolutional network (GNN-ResGated), which were used as baselines. Furthermore, GNN-SAGE generated smaller errors than those reported in the current literature. The Shapley additive explanations (SHAP) revealed that prior data from the reference station was the most significant factor for all prediction intervals, with seasonality and precipitation being more influential for longer-range forecasts. The findings positioned the proposed GNN-SAGE model as a cutting-edge solution for flood forecasting and a valuable resource for devising early flood-warning systems.

**Keywords:** flooding; Humber River; forecasting; machine learning; graph neural networks; SHAP analysis

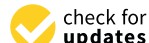



## 1. Introduction

One of global warming's adverse effects is the increase in the frequency and severity of floods [1,2]. Resulting from both anthropologic and weather factors [3–7], floods are the deadliest natural event [8,9], where they have already affected billions of people worldwide [10–12], causing huge economic and human losses [13–17]. In Canada, flood events can happen all year and are caused by snow melting and thunderstorms [18,19]. The frequency and intensity of floods have been increasing in recent years [10,20]; they were responsible for around 32% of the total natural disasters between 1950 and 2012 [1,21] and accounted for nearly 75% of the Canadian Disaster Financial Assistance Arrangements budget [18,22], pressuring municipalities to improve their infrastructure and management strategies to accommodate these extreme events [12,17,23].

To diminish flood impacts and help decision-making and proper management by authorities, researchers directed their efforts toward early flood detection models, allowing for the deployment of pre-emptive countermeasures and mitigating adverse flooding effects [19]. To this end, many predictive models were developed, which are divided into two major groups [24]: hydrological and data-driven intelligent models. Hydrological models rely on physics-based equations to determine future flood occurrences. Even though this approach achieved good results in previous studies [25–29], it has limitations

regarding its complex modeling, computational cost, the need for many hydro-geomorphic input attributes, and high precision of mapping attributes, which may result in large errors [10,19,30–32].

To overcome the physics-based models' limitations, data-driven models were employed using machine learning (ML) approaches. Researchers studied this paradigm because ML models are simpler, require fewer input parameters, have superior processing times, can solve non-linear relations, and identify the complex relationships between input and output parameters [5,24,33,34].

In [35], an ML model based on the autoregressive integrated moving average (ARIMA) was implemented to assess flooding of the Jhelum River, which is located in the Great Himalayas region. The authors used 41 years of maximum peak flow data to build the model and attained satisfactory forecasting results, deeming it suitable for predicting flood events in the region. In another work [36], the authors used an artificial neural network (ANN) multilayer perceptron (MLP) model to forecast floods in Southern Thailand. The researchers fused meteorological, hydrological, geospatial, and big data. The results showed that their proposed approach attained the best values when compared with other ML models, achieving 97.83% accuracy for flood forecasting. Another recent ML application was based on the group method of data handling (GMDH) [37,38]. A flood forecasting model based on this paradigm was first proposed by [39] for a study set in Iowa, where the proposed model achieved the best results when compared with traditional models, reaching a root-mean-squared error (RMSE) of 42.51 $m^3/s$. Later studies acknowledged this approach's superior performance when compared against conventional forecasting paradigms, such as ARIMA [1], with it being suited to forecast riverine flooding events up to 17 h in advance [10].

Data-driven models can use satellite information to calibrate and model river discharges [40,41] and can be used together with ML techniques for improved forecasting performance. In [42], a hybrid model using satellite information and ML techniques was implemented for forecasting floods in the USA. In their work, the authors successfully corrected faulty satellite measurements, increasing the accuracy for the determination of future flood events. In another work [43], the authors used geophysical and anthropogenic inputs to assess flood-susceptible areas in Iran. In their work, flood events were assessed using random forests (RFs) and a Bayesian generalized model, where the former showed better accuracy for this task. Another work [44] implemented Markov chains and Monte Carlo uncertainty analysis over satellite signals to improve flooding forecasting. The results demonstrated that the author's proposed approach surpassed the traditional ML models that were used as benchmarks.

Models based on deep neural networks (DNNs) are an improvement over the traditional ML paradigms [45]. Le et al. [46] used a long short-term memory (LSTM) DNN model for flooding forecasting in Vietnam up to three days ahead. The best results were reached when upstream and study stations' discharge data were used, achieving the optimal RMSE values of 151 $m^3/s$, 373 $m^3/s$, and 594 $m^3/s$ for one, two, and three days ahead, respectively, showing this method to be a reliable tool for flood modeling. Besides LSTM, convolutional neural networks (CNNs) based models were evaluated by Kabir et al. [47]. In their work, the implemented CNN model was trained with 2D data from a hydraulic model to forecast floods in the United Kingdom for 2005 and 2015. Compared with the benchmark model, CNN showed better computational performance and higher precision for such tasks.

Another viable approach for DNN models is combining them, resulting in a hybrid model. These models can reach better forecasting results in time-series problems [33,48,49]. A hybrid model was developed by [19], where an attention layer was embedded in the LSTM model. The proposed model sought to establish a spatiotemporal linkage between the neighboring stations, resulting in a more accurate model for flooding prediction across Canada. Their proposed spatiotemporal attention LSTM (STA-LSTM), when compared against CNN-LSTM and another LSTM-based model, was found to attain improved fore-

casting results, with the lowest error rate of 3.98%. A CNN-LSTM hybrid approach was proposed in [50] for river flow forecasting in Germany and possible flooding events. The authors used the hybrid model to process 2D radar imagery, processing spatial and temporal information using CNN and LSTM, respectively. Their results showed that the CNN-LSTM model attained overall satisfactory results for high-water and low-water discharge periods, which are useful for early flooding alert systems.

However, traditional DNN models cannot accurately process multi-dimensional spatiotemporal data information [51,52]. To address this hindrance, graph-based models were used for spatiotemporal forecasting problems in different time-series studies, such as weather prediction [51], pollution level estimation [53], and wind speed forecasting [54,55]. The use of a graph-based model for flooding forecasting was analyzed by Feng et al. [56] for a location in China. In their work, a graph convolutional network (GCN) was implemented with LSTM to better extract the spatiotemporal information underlying the input data. The proposed model was compared against six benchmark ML models, and it was found to achieve the best RMSE results for all forecasting horizons, with an average of 84.76 m$^3$/s.

The literature review given above shows the importance of data-driven models using DNN and recently graph-based approaches for forecasting flooding events. To deepen the understanding of flooding prediction, the present work proposed a novel forecasting graph-based model to determine the flooding occurrence in Humber River, Ontario, Canada. The model aimed to contribute to the field in the following ways:

1.  Develop a new state-of-the-art model for flooding forecasting, allowing for a more precise and accurate early flooding alert system.
2.  Verify the addition of spatiotemporal data for improved flooding forecasting results.
3.  Development of a reliable ML model based on graph theory and DL paradigm.

The remainder of this work is divided as follows: in Section 2, the methodology used is presented, followed by Section 3, where the achieved results are shown. In Section 4, there is a discussion of the results, and Section 5 closes the work with a conclusion.

## 2. Proposed Model

### 2.1. Persistence Model

The persistence model was selected as a benchmark for the present work to compare the results with the proposed graph-based deep learning model. The persistence model is widely used as a baseline for forecasting algorithms applied to intermittent and non-linear time series. This model is a hard contender to be surpassed for short forecasting horizons [57,58], offering good results by simply stating that the future estimated value will be the same as the one measured in the present [59,60]. Nevertheless, as the horizon extends, the persistence model fails to capture the dynamicity of the trends for seasonality and external factors that may influence the estimated value, reducing its performance [61].

### 2.2. GNN-SAGE and GNN-ResGated Models

The present work proposed a new graph neural network sample and aggregate (GNN-SAGE) model for forecasting potential flooding occurrences. This model, as its name suggests, uses graph theory to retrieve spatiotemporal information from the dataset. The GNN-SAGE framework extracts spatiotemporal information by equally sampling and aggregating its nodes, with it being especially efficient in handling large graphs.

During the sampling phase, several neighboring nodes are selected rather than using all available nodes. This predefined selection improves the computational time at the expense of reduced information being given to the model. In the following phase, an aggregator is implemented to collect information from the previously selected neighboring nodes, resulting in an embedding vector that represents the node of interest that can generalize unknown information apart from the graph shape [53,62–65].

The present study used the GNN-SAGE model to predict future stage levels, employing spatiotemporal information from neighboring measuring stations in the Humber River. Each station, which provided both precipitation and stage information, contributed spa-

tiotemporal data for forecasting the stage level at the reference station, which was located near the river's estuary. This information was fed into the GNN-SAGE model as input data, effectively capturing the spatial and temporal relationships between the measuring stations. Its inherent capacity to identify and retrieve the underlying spatiotemporal information within the dataset was an important aspect of the GNN-SAGE model for the present application due to both spatial and temporal features being major drivers of flooding occurrence. The following Figure 1 depicts the GNN-SAGE framework used in this study.

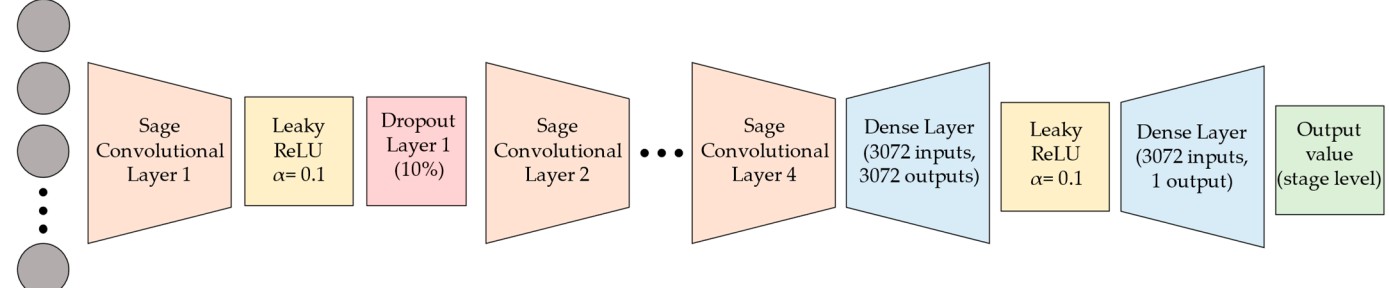

**Figure 1.** General GNN-SAGE architecture structure used.

Figure 1 shows the framework for the GNN-SAGE model: the spatiotemporal data from the stations (represented as graph nodes) are the input information. After this, there is a succession of convolutional, activation, dropout, and dense layers, identifying patterns and extracting local features and spatiotemporal information as the inputs go deeper into the model.

The graph-based model residual gated graph convolutional network (GNN-ResGated) was also selected to benchmark the proposed model's performance. The authors Bresson and Laurent [66] originally proposed GNN-ResGated as an improvement over the works of He et al. [67] and Marcheggiani and Titov [68]. The GNN-ResGated combines graph-based models and long short-term memory (LSTM). It differs from GNN-SAGE in how it processes its data: while GNN-SAGE operates by sampling and extracting data from each node, the GNN-ResGated approach considers residual connections and recurrent information between the graph's layers [69]. In the GNN-ResGated architecture, the residuality allows for stacking multiple layers within the model, improving the spatial data extraction [70], while the LSTM controls the data flow and processes the temporal aspect of the data, also deciding what information can be kept or discarded. This approach was found to have superior performance over other graph and recurrent neural network models [66].

### 2.3. SHAP Analysis

Despite its attested performance and state-of-the-art results in many research fields, ML models still lack explainability, not offering a complete understanding of their outcomes. The Shapley additive explanations (SHAP) (documentation available at https://shap.readthedocs.io/en/latest/, accessed on 5 April 2023) uses game theory to provide further insight into the ML results, relating input variables and the achieved output value in terms of importance, correlation, and influence of each input variable over the final prediction [53,71]. The SHAP analysis was found to be a reliable tool for authors to fully understand their results in multidisciplinary fields, such as pharmaceuticals [72], material engineering [73], Earth system modeling [74], and social factors driving income [75].

### 3. Validation and Analysis of Results

#### 3.1. Humber River Description

The Humber River is located in the southern part of the Ontario province in Canada, with it being Lake Ontario's most important tributary river. The Humber River drainage

area is approximately 903 km², comprising an estimated population of around 850,000. Its land use is mixed between rural and urban areas in the upstream area but becomes increasingly urbanized as it approaches Toronto [19,76,77]. The increase in the urbanized area leads to an increase in the amount of runoff after rains due to the impervious surface, resulting in a region prone to flooding [76,78]. In past years, this region has suffered from major floods, causing both economic damage and the loss of human lives.

In the present study, both stage and precipitation data were used to assess the occurrence of flooding. The historical data were from the period 8 June 2018 to 17 September 2020, with a time resolution of 15 min for stage data and 5 min for precipitation data. It was retrieved from The Toronto and Region Conservation Authority (available at https://backup.trcagauging.ca/, accessed on 30 March 2023). Figure 2 depicts the stations used using stage data and precipitation data, and the southernmost red symbol represents the reference station, where we wanted to determine flooding occurrences by forecasting the river stage at this location.

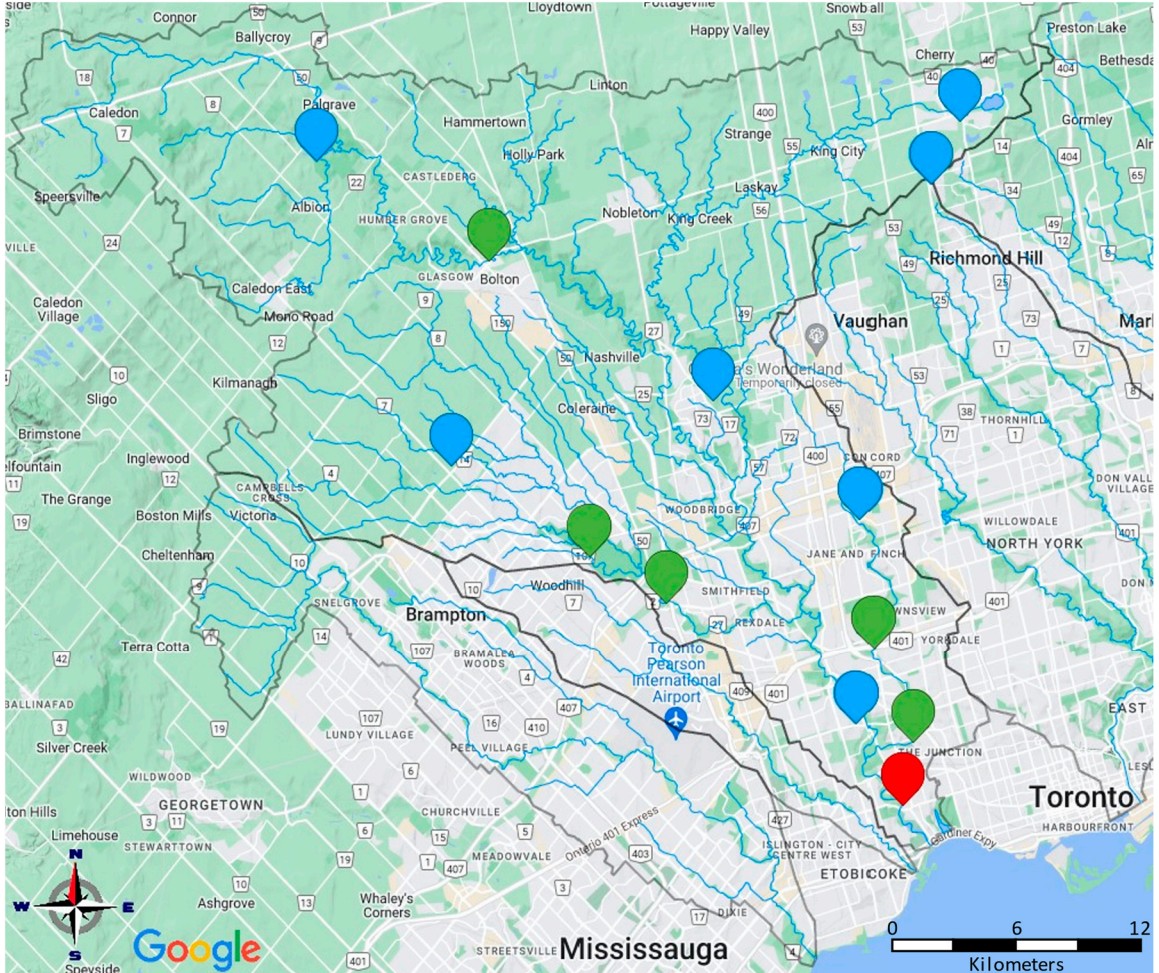

**Figure 2.** Map showing the location of the stations containing stage (in green) and precipitation (in blue) data. The red symbol is the site of the reference station.

### 3.2. Evaluation Metrics

The results forecasted with GNN-SAGE were compared with the benchmarking models using the metrics root-mean-squared error (RMSE), mean average error (MAE), mean absolute percentage error (MAPE), and coefficient of determination ($R^2$). Their equations can be found in [79] for $R^2$ and [60] for the remainder.

### *3.3. Dataset Size Evaluation*

The training dataset was tested to verify whether the used dataset size could provide stable results when the proposed model was used to forecast the river stage. Figure 3 depicts the results without (only stage data) and with precipitation data.

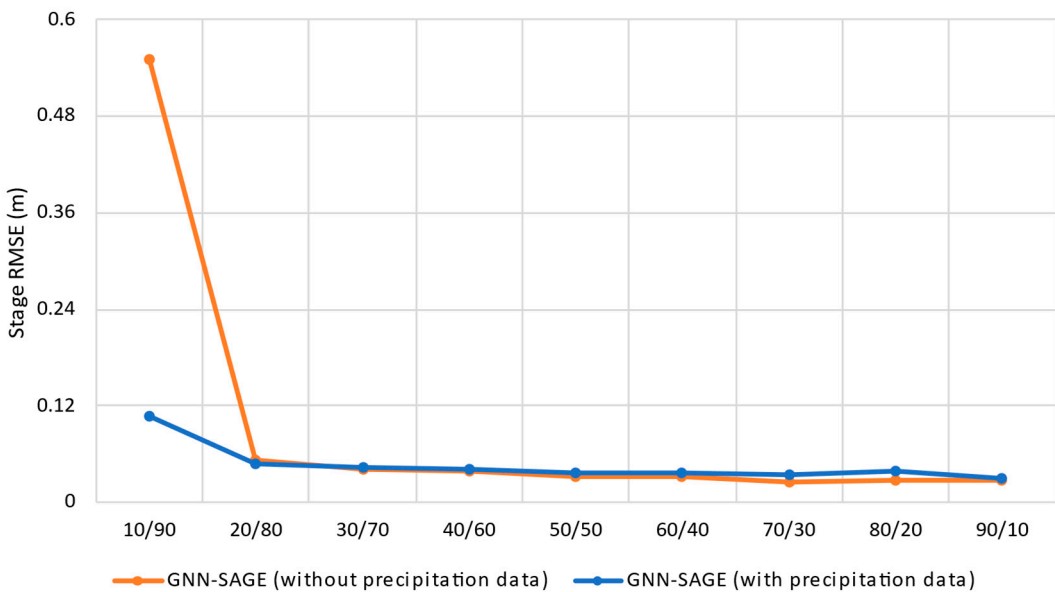

**Figure 3.** Influence of the different dataset sizes without and with precipitation data, as represented by the colors orange and blue, respectively.

In Figure 3, the y-axis represents the RMSE error for the target variable stage in meters, while the x-axis shows the ratio of the used data for evaluating the model. Figure 3 shows that for each assessed dataset size, increasing the training data size provided better results in terms of reduced RMSE values, showing convergence when a ratio greater than 10/90 was used for both cases with and without precipitation data. The optimal configuration for GNN-SAGE without the precipitation information was reached for a ratio of 70/30, where the model's RMSE was 0.0291 m. For the case where the dataset considered precipitation data, the best result was found at 90/10, with an RMSE equal to 0.3057 m. Ultimately, the data was divided using an 80/20 Pareto ratio, retaining more information than a 70/30 split while avoiding the limited data concerns associated with a 90/10 distribution.

### *3.4. Results for a 1 h Forecast Horizon*

Multiple time lag values were evaluated for predicting floods 1 h in advance. To compare the proposed GNN-SAGE model, its results were analyzed against persistence and GNN-ResGated models. The results are presented in Figure 4, where the x-axis represents the time lag in hours.

Figure 4 shows that for 1 h forecasting, the GNN-ResGated model had no improvement over its performance for time lags greater than 6 h, while for GNN-SAGE, it improved the overall model's performance consistently by a slight margin after a 3 h time lag. The proposed GNN-SAGE achieved the best results for all time lags used, improving the flooding forecasting by up to 9.95% when compared with the persistence model and 18.93% on average when compared with GNN-ResGated. Conversely, GNN-ResGated did not achieve satisfactory results, with it being surpassed by the other two models for all considered time lags, and it was not able to capture the spatiotemporal characteristics of the dataset satisfactorily. For GNN-SAGE, it was deemed that a 6 h time lag was sufficient for the model's prediction performance since it showed convergence after a 3 h time lag.

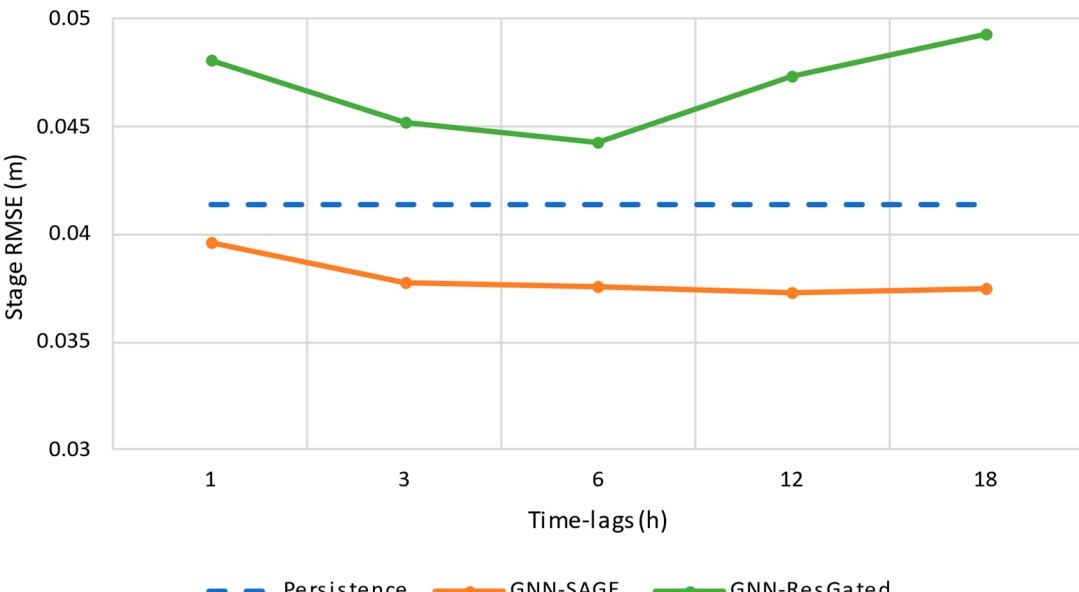

**Figure 4.** Influence of the time lag on the GNN-SAGE and GNN-ResGated models for a 1 h forecasting horizon compared with persistence.

Different input variables were tested for the GNN-SAGE and GNN-ResGated models for 1 h forecasting, as depicted in Figure 5, where its x-axis represents the used variables (day of year, hour of day, precipitation, and stage level).

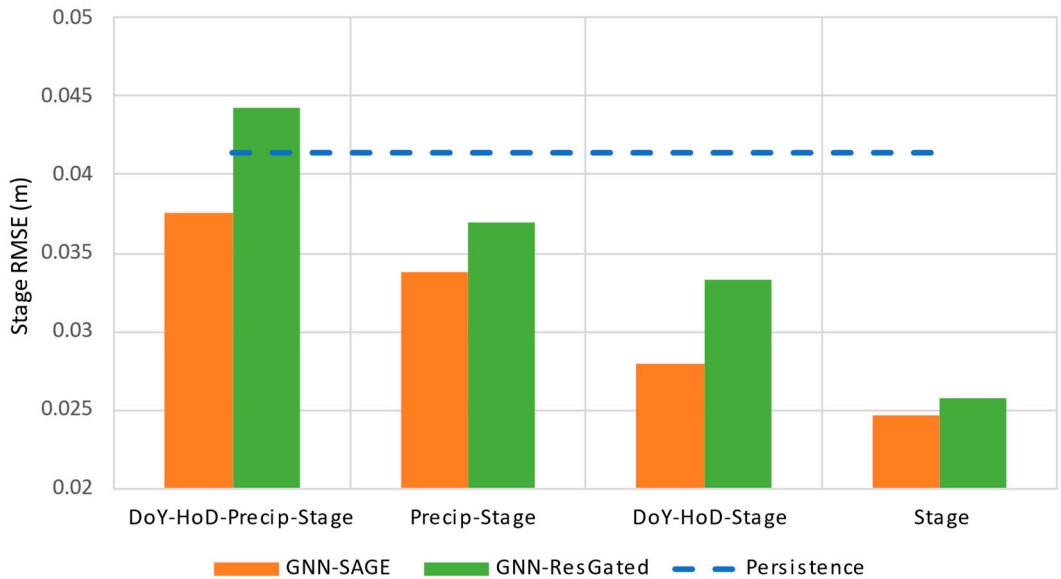

**Figure 5.** Influence of the different input variables for the GNN-SAGE and GNN-ResGated models for a 1 h forecasting horizon.

Figure 5 demonstrates that, in terms of understanding the hydrological response, the best performance for both models was achieved when using 6 h of past data solely for the input variable "Stage", with errors equal to 0.02466 m (40.36% improvement over the persistence model) and 0.02574 m (37.76% improvement over the persistence model) for GNN-SAGE and GNN-ResGated, respectively. This result indicates that the DoY, HoD, and precipitation data from neighboring upstream stations did not provide meaningful information for the model, and thus, did not exert a positive impact over its forecasting for a 1 h horizon using past data from up to 6 h.

Figure 6 depicts the accuracy of the GNN-SAGE model forecasting compared with the real measured stage values for the Humber River validation dataset.

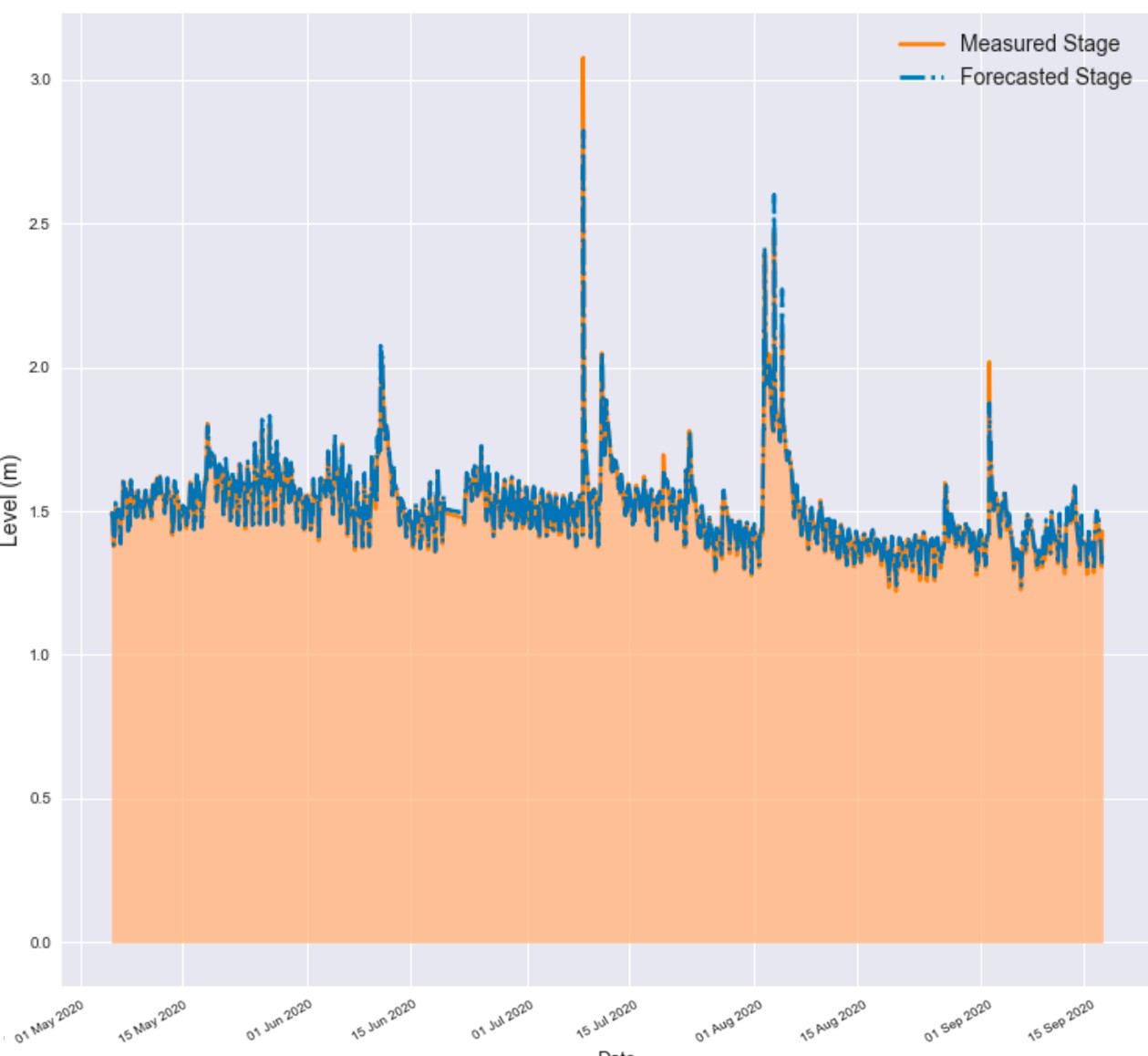

**Figure 6.** Comparison between the forecasted stage values using GNN-SAGE and real measured stage values for Humber River for a 1 h forecasting horizon.

From Figure 6, it is possible to see that the predicted stage values closely followed the real ones. It is important to highlight that GNN-SAGE managed to identify all the peaks for the assessed period, though slightly underestimating them. This is critical in identifying possible flooding events where the stage levels of the river surpass the safety limits for a location. The exceptional correlation between the predicted and actual stage values was evident in the regression line for the model, as illustrated in Figure 7. The associated histograms display the frequency of the statistical probability distribution for the stage values.

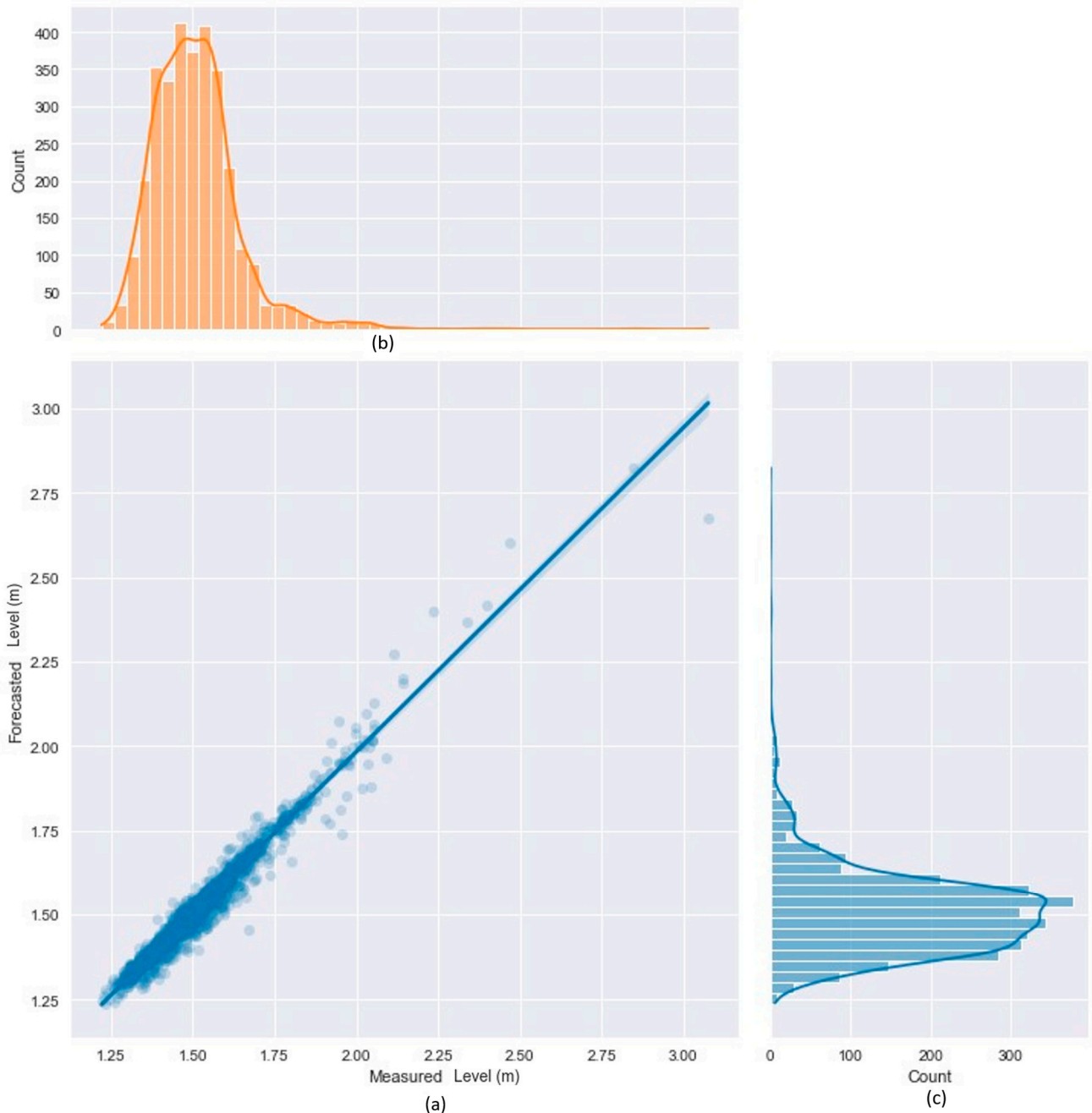

**Figure 7.** Scatter plot (**a**) with the marginal distribution for the measured (**b**) and forecasted (**c**) level values using GNN-SAGE for 1 h ahead.

Figure 7 confirms the concurrence between the predicted and actual stage values, as demonstrated by the clustering of points around the regression line, with a higher concentration between 1.25 m and 1.75 m for both the forecasted and real values. For this scenario, $R^2$ reached an excellent value of 96.45%.

### 3.5. Results for a 3 h Forecast Horizon

The following results refer to flood forecasting 3 h in advance. Figure 8 shows the effects of different time lags on the models' performances.

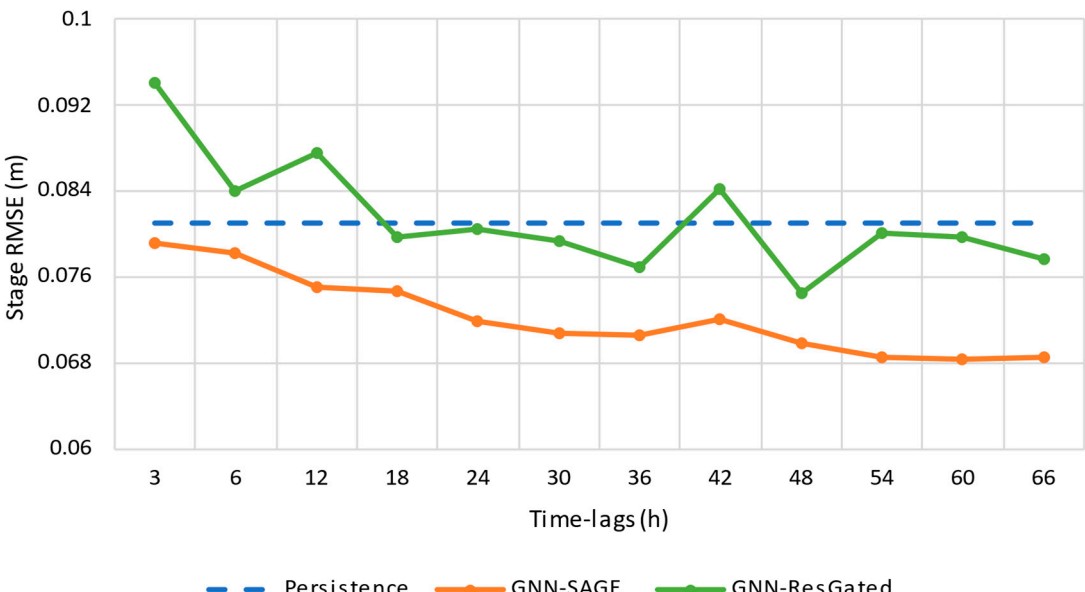

**Figure 8.** Influence of the different time lags for the GNN-SAGE and GNN-ResGated models for a 3 h forecasting horizon compared with the persistence model.

For a 3 h forecasting horizon, again, GNN-SAGE achieved the best results, outperforming both the persistence model and GNN-ResGated for all assessed time lag windows. This time, both models benefited from previous information, showing a decreasing error trend: GNN-ResGated improved its forecasting by 8.11% compared with the persistence model for a 48 h time lag, with an RMSE equal to 0.07451 m. Furthermore, GNN-SAGE managed to improve the flooding forecasting by up to 15.87% and 14.36% when compared with the persistence and GNN-ResGated models, respectively, for the optimal time lag of 60 h and with an RMSE equal to 0.06822 m. Concerning GNN-SAGE, after a 60 h time lag, the model showed convergence in its results, thus deeming this number enough to return satisfactory results for stage forecasting.

Figure 9 shows the results for different input variables.

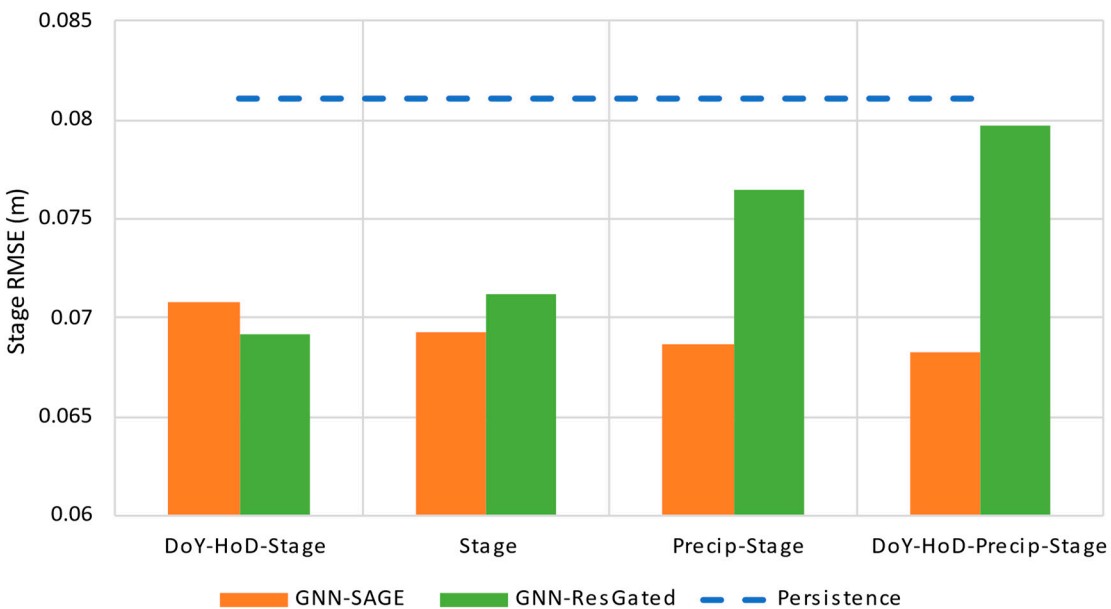

**Figure 9.** Influence of the different input variables for the GNN-SAGE and GNN-ResGated models for a 3 h forecasting horizon.

The best performance for the 3 h forecasting horizon for the proposed GNN-SAGE was achieved using past information from all input variables and DoY/HoD. For this input data configuration, the GNN-SAGE RMSE was equal to 0.06822 m, achieving improvements of 15.87% over the persistence model and 14.36% over GNN-ResGated. Figure 9 indicates that the 3 h forecasting horizon benefits from the DoY, HoD, and precipitation information that reached the reference station positively influenced the model's performance by adding significant temporal information from neighboring stations.

Since precipitation data was used for a 3 h forecasting configuration, verifying the precipitation influence over the model's performance was important. To do so, the precipitation summing window, which indicated previous accumulated precipitation levels, was assessed for different sizes, ranging from 1 h to 24 h in the past, as presented in Figure 10.

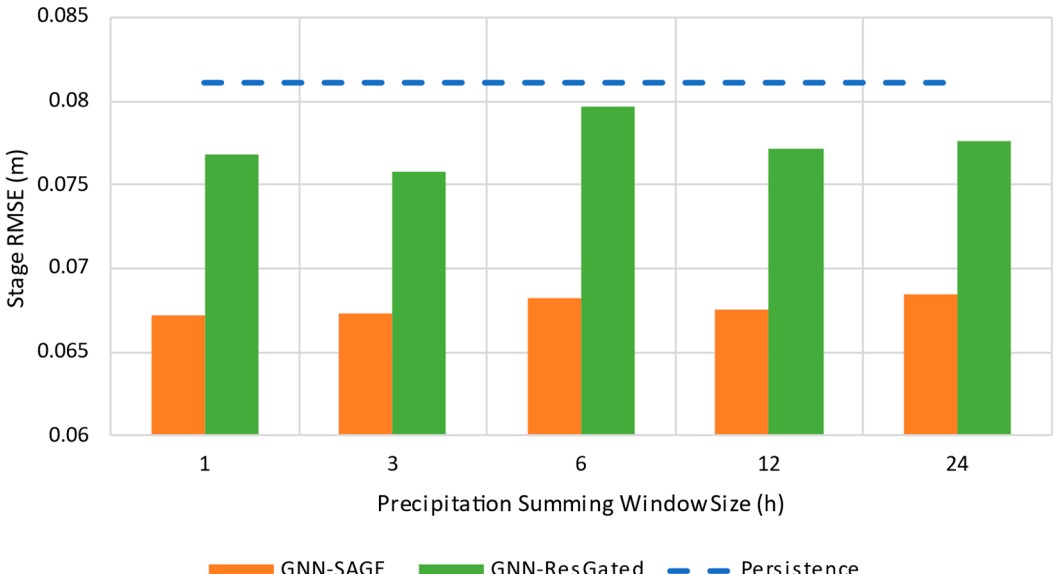

**Figure 10.** Influence of the different precipitation summing window sizes for a 3 h forecasting horizon.

The best error value was reached using a window size of 1 h, with an RMSE equal to 0.06721 m, indicating improvements of 17.11% and 12.51% over the persistence model and GNN-ResGated, respectively. This means that the total accumulated rain from the past 1 h was enough to provide information for the GNN-SAGE model regarding the stage level prediction. Figure 11 shows the forecasted stage values obtained using the proposed model, comparing them with the real measured values over a reference period.

Similarly to Figure 6, the proposed graph model followed the real stage values but was less accurate than before. Concerning the level peaks for the reference period, the model underestimated them, especially the highest peak over 3 m, which occurred between 1 July and 15 July. As expected, this reflected an inferior performance compared with the previous 1 h forecasting horizon, as illustrated in Figure 12.

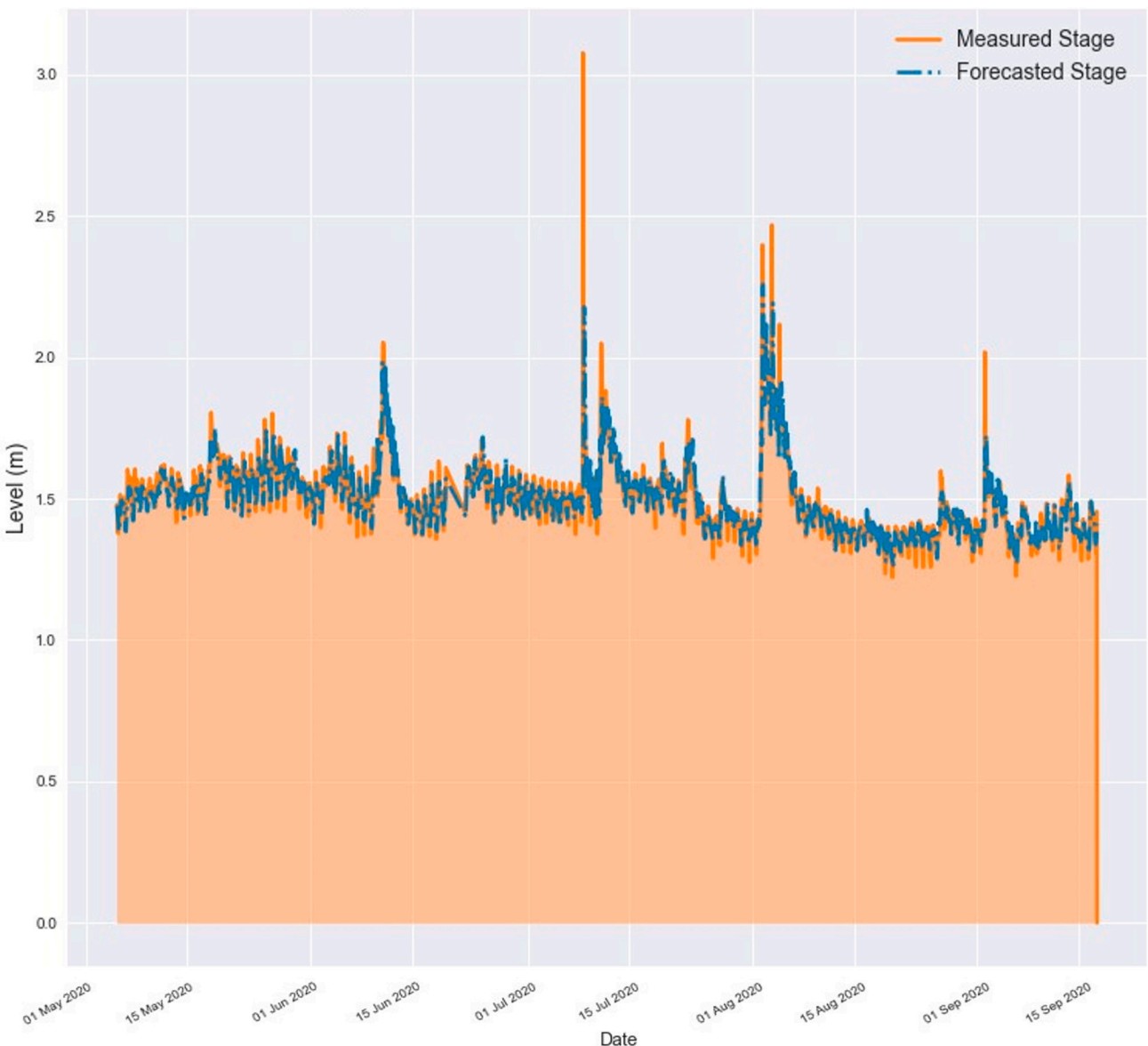

**Figure 11.** Comparison between forecasted stage values using GNN-SAGE and real measured stage values for Humber River for a 3 h forecasting horizon.

For a 3 h forecasting horizon, there was still a good agreement between the forecasted and real data. There was more scattering around the regression line, even though the GNN-SAGE model still returned good forecasting values. The measured values were located within the same interval of 1.25 m and 1.75 m, and the forecasted data were located between 1.20 m and 1.80 m, resulting in a variance higher than for the 1 h forecasting horizon situation. The resultant $R^2$ for this case was 75.49%.

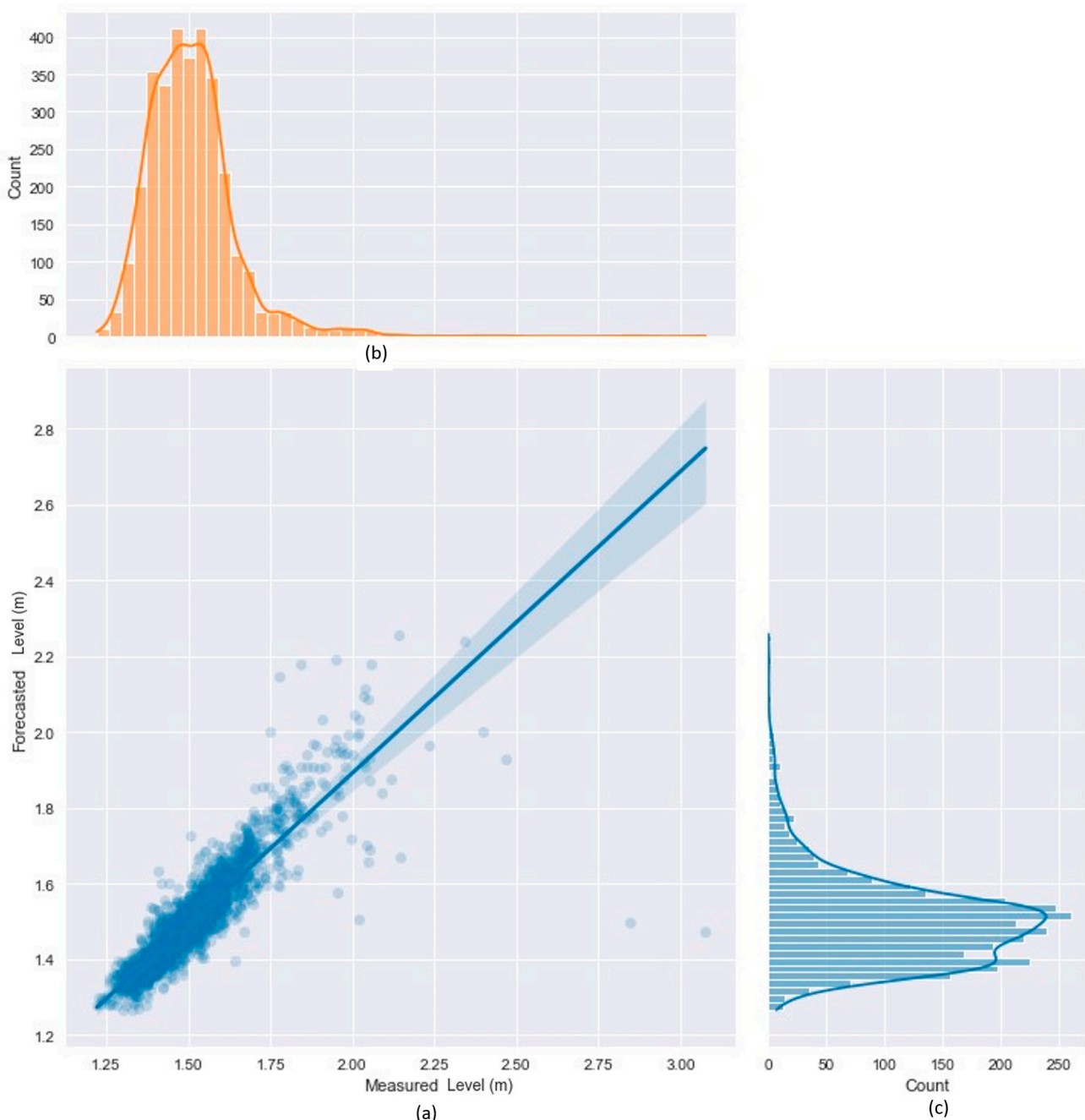

**Figure 12.** Scatter plot (**a**) with a marginal distribution for the measured (**b**) and forecasted (**c**) level values using GNN-SAGE for 3 h ahead.

### 3.6. Results for a 6 h Forecast Horizon

Considering a forecasting horizon of 6 h, the results found for the effect of different time lags over the models' performances are shown in Figure 13.

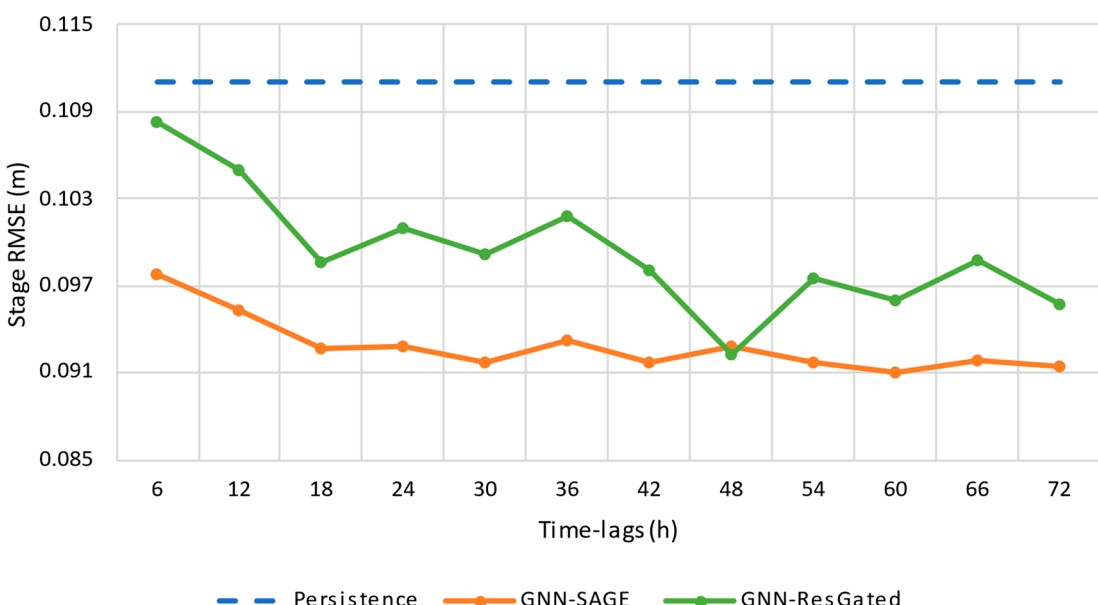

**Figure 13.** Influence of the different time lags for the GNN-SAGE and GNN-ResGated models for a 6 h forecasting horizon compared with the persistence model.

For the 6 h forecasting horizon, both GNN-SAGE and GNN-ResGated managed to surpass the reference model for all studied time lags: the former attained the best overall results for a 60 h time lag with an RMSE of 0.09103 m; meanwhile, the latter achieved better results for a 48 h time lag with an RMSE equal to 0.09227 m. Again, GNN-SAGE was the best model for the analyzed time lag interval, improving the stage forecasting by up to 17.97% compared with the persistence model and 5.19% compared with GNN-ResGated. Figure 14 presents the results for different input variables.

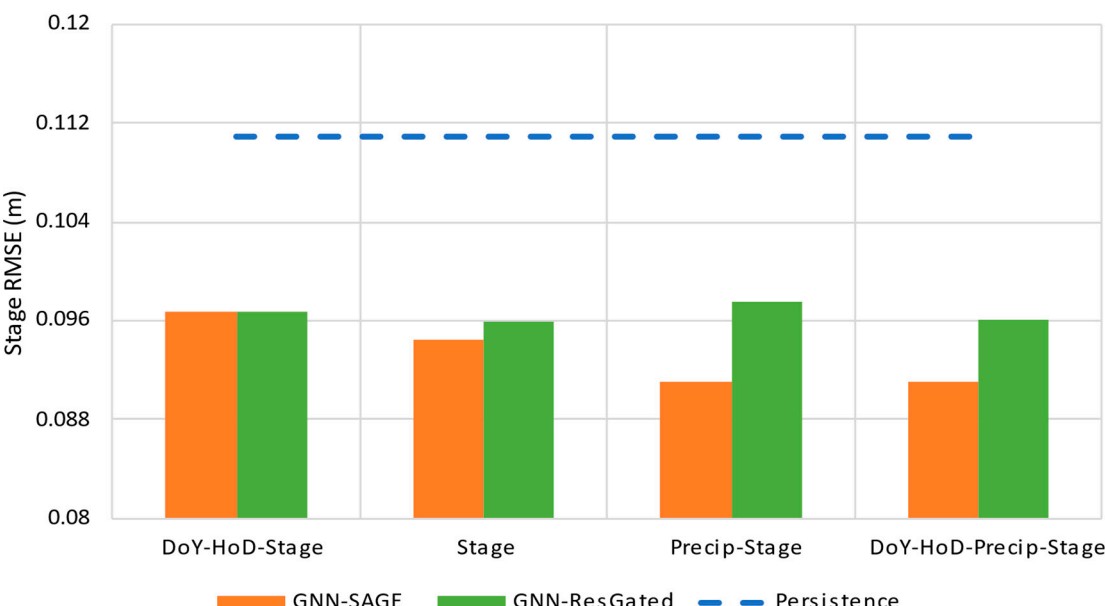

**Figure 14.** Influence of the different input variables for the GNN-SAGE and GNN-ResGated models for a 6 h forecasting horizon.

As expected, due to the large time lag value of 60 h, all input variables contributed positively to GNN-SAGE performance since there was time for them to reach the reference station downstream. For this configuration, the best error value for the proposed model was 0.09103 m. Interestingly, both graph-based models achieved the same error values when

DoY, HoD, and stage were used together as input variables, indicating that using these data together did not contribute to improving the models' performance. In this situation, adding precipitation data was fundamental for the GNN-SAGE performance and was achieved by supplementing extra spatiotemporal information to the model. For GNN-ResGated, the model presented almost the same error values for all assessed cases, indicating that the model could not correctly extract spatiotemporal information from the data, which did not improve its forecasting capacity. Figure 15 presents the models' behavior regarding the accumulated precipitation data for different intervals in terms of the window size in hours.

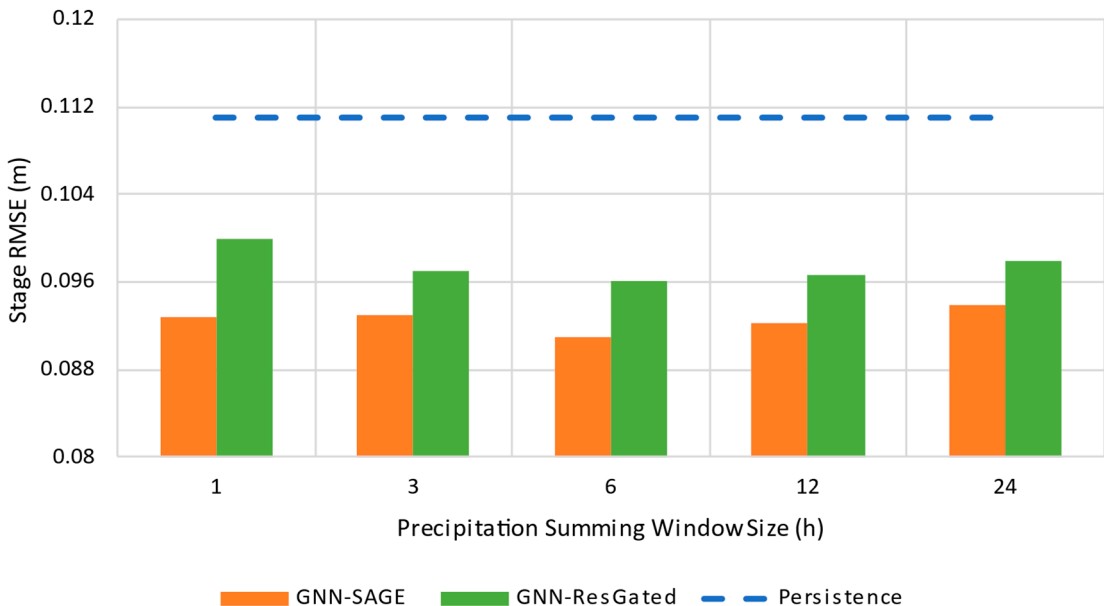

**Figure 15.** Influence of the different precipitation summing window sizes for a 6 h forecasting horizon.

Concerning 6 h forecasting, the total accumulated precipitation in the past 6 h provided sufficient information for the GNN-SAGE model. For this time window, the proposed model error was 0.09103 m, the same as that presented in Figure 13, and this model achieved the same improvements compared with the other two models. In Figure 16, a comparison between the forecasted and validation values is presented.

For the 6 h forecasting horizon, the model somewhat followed the actual data trend, though it significantly missed the peaks. This behavior is typical for predictive models that forecast values too far into the future, thus losing the connection between the data, resulting in averaged forecasted values [53]. For this scenario, the peaks were significantly underestimated, resulting in an inferior performance, as pictured in Figure 17.

Figure 17 shows more scattering than the previous scatter plots for 1 h and 3 h forecasting horizons. The forecasted values were located more sparsely between 1.20 m and 1.80 m. This higher variance around the regression line reflected the proposed model's performance, resulting in a reduced $R^2$ with a value of 59.07%.

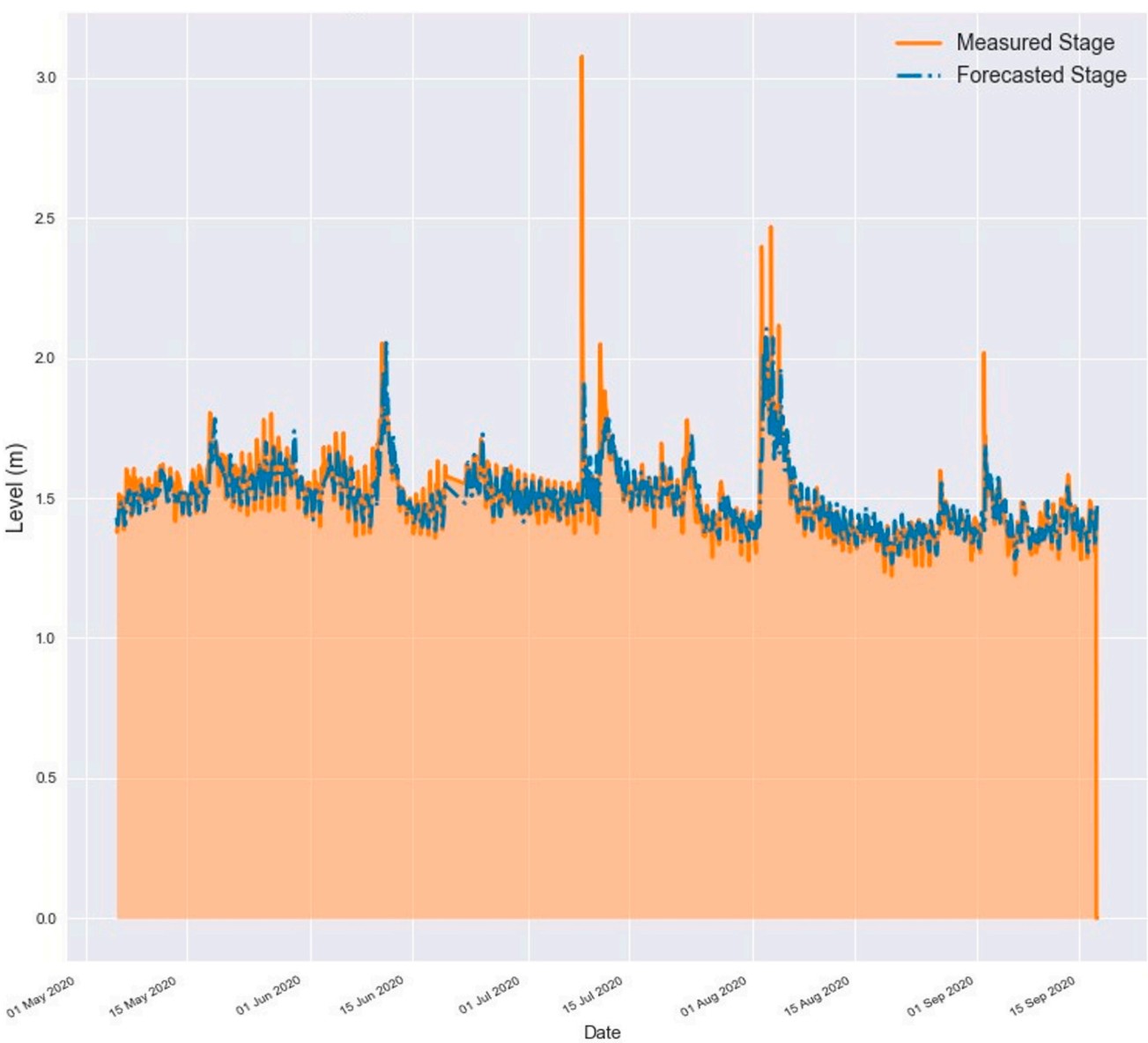

**Figure 16.** Comparison between the forecasted stage values using GNN-SAGE and the measured stage values for Humber River for a 6 h forecasting horizon.

### 3.7. Results for a 12 h Forecast Horizon

Considering a forecasting horizon of 12 h, the results found for the effects of different time lags on the models' performances are shown in Figure 18.

Figure 18 points out that both models could improve their forecasting performances when data from further past was added, surpassing the persistence model for all tested time lags. For GNN-SAGE, the model started to present convergent behavior after a 66 h time lag and had the best configuration for the 12 h forecasting horizon situation among all models, with the minimum error value of 0.12293 m. Compared with the persistence model and GNN-ResGated, for the same time lag, the proposed model improved the forecasting performance by 13.33% and 6.69%, respectively. The following Figure 19 shows the different input data used for each model.

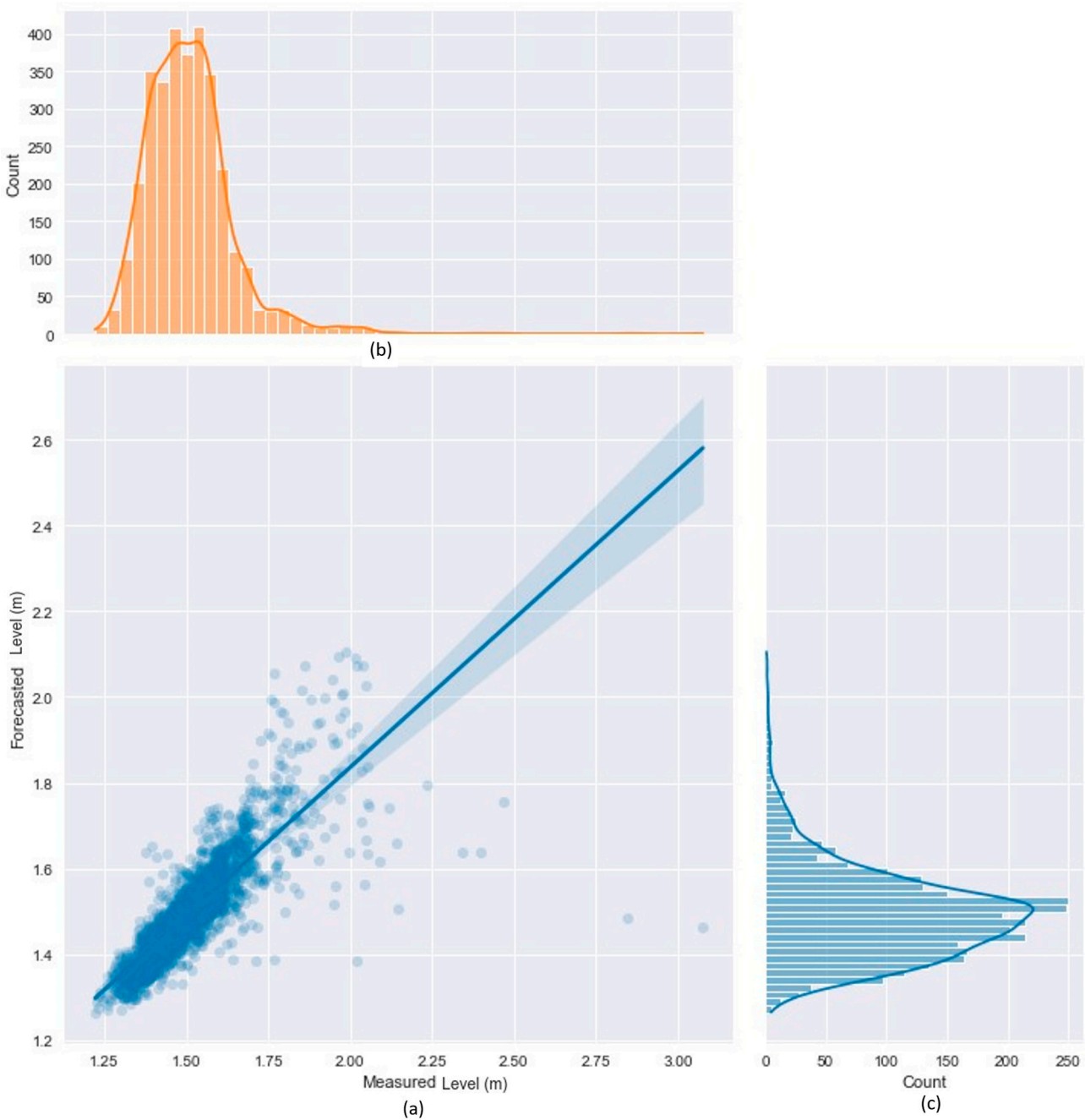

**Figure 17.** Scatter plot (**a**) with a marginal distribution for the measured (**b**) and forecasted (**c**) level values using GNN-SAGE for 6 h ahead.

Again, for a 12 h forecast, using all variables information was found to increase GNN-SAGE's performance, resulting in the lowest RMSE value of 0.12293 m. Using the variable stage alone resulted in similar errors for both models, and the case using DoY, HoD, and stage. Similar to the 6 h forecasting horizon scenario, the addition of extra precipitation information could improve GNN-SAGE's performance. Figure 20 shows the accumulated precipitation data for different time intervals in the past.

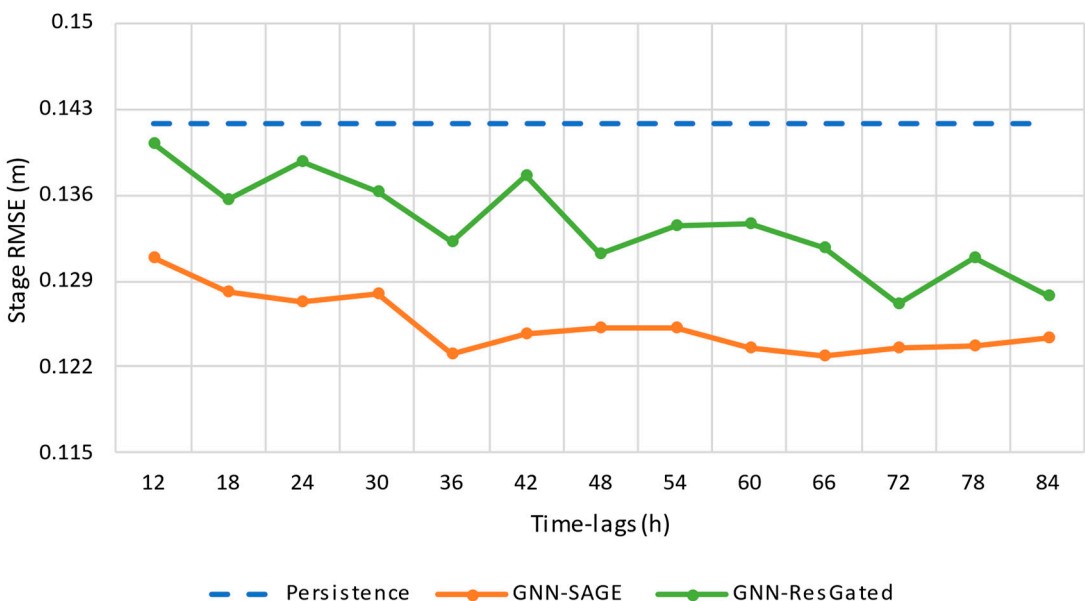

**Figure 18.** Influence of the different time lags for the GNN-SAGE and GNN-ResGated models for a 12 h forecasting horizon compared with the persistence model.

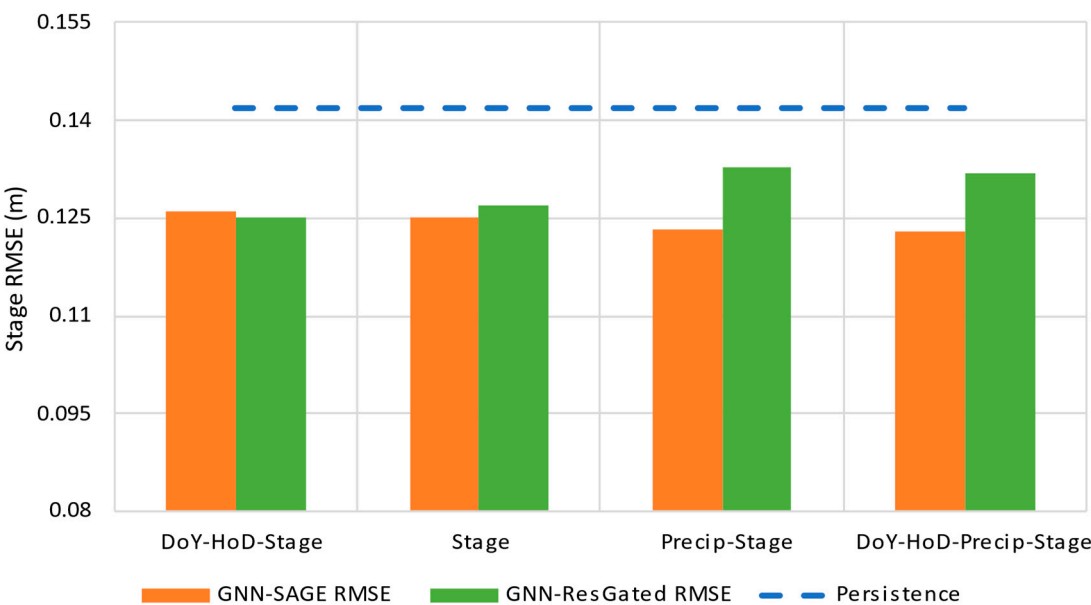

**Figure 19.** Influence of the different input variables for the GNN-SAGE and GNN-ResGated models for a 12 h forecasting horizon.

For the assessed case forecasted 12 h in advance, accumulated rain from the last 3 h sufficed for the model forecasting performance. Using this summing window size, the RMSE was equal to 0.12227 m, providing an increase of 13.79% over the persistence model and a 6.97% increase over the remaining GNN-ResGated model. Figure 21 illustrates how well the forecasted values followed the measured values' trend over the reference period.

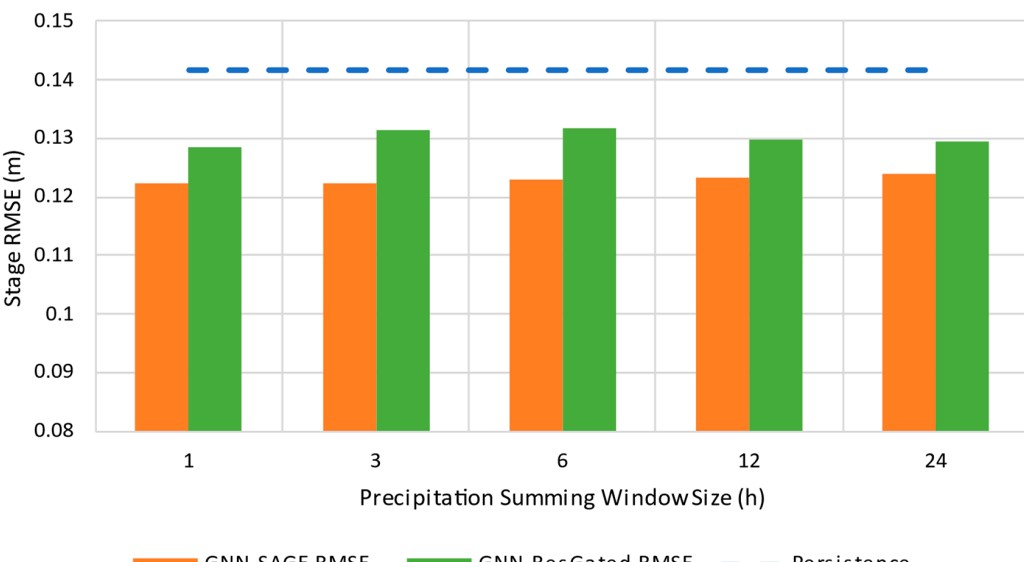

**Figure 20.** Influence of the different precipitation summing window sizes for a 12 h forecasting horizon.

**Figure 21.** Comparison between the forecasted stage values using GNN-SAGE and the real measured stage values for Humber River for a 12 h forecasting horizon.

As expected, the model's performance deteriorated for an increased forecasting horizon of 12 h. Although GNN-SAGE still followed the data trend, most of the peaks were missed by its predictions. This reduced performance is better visualized in the following Figure 22.

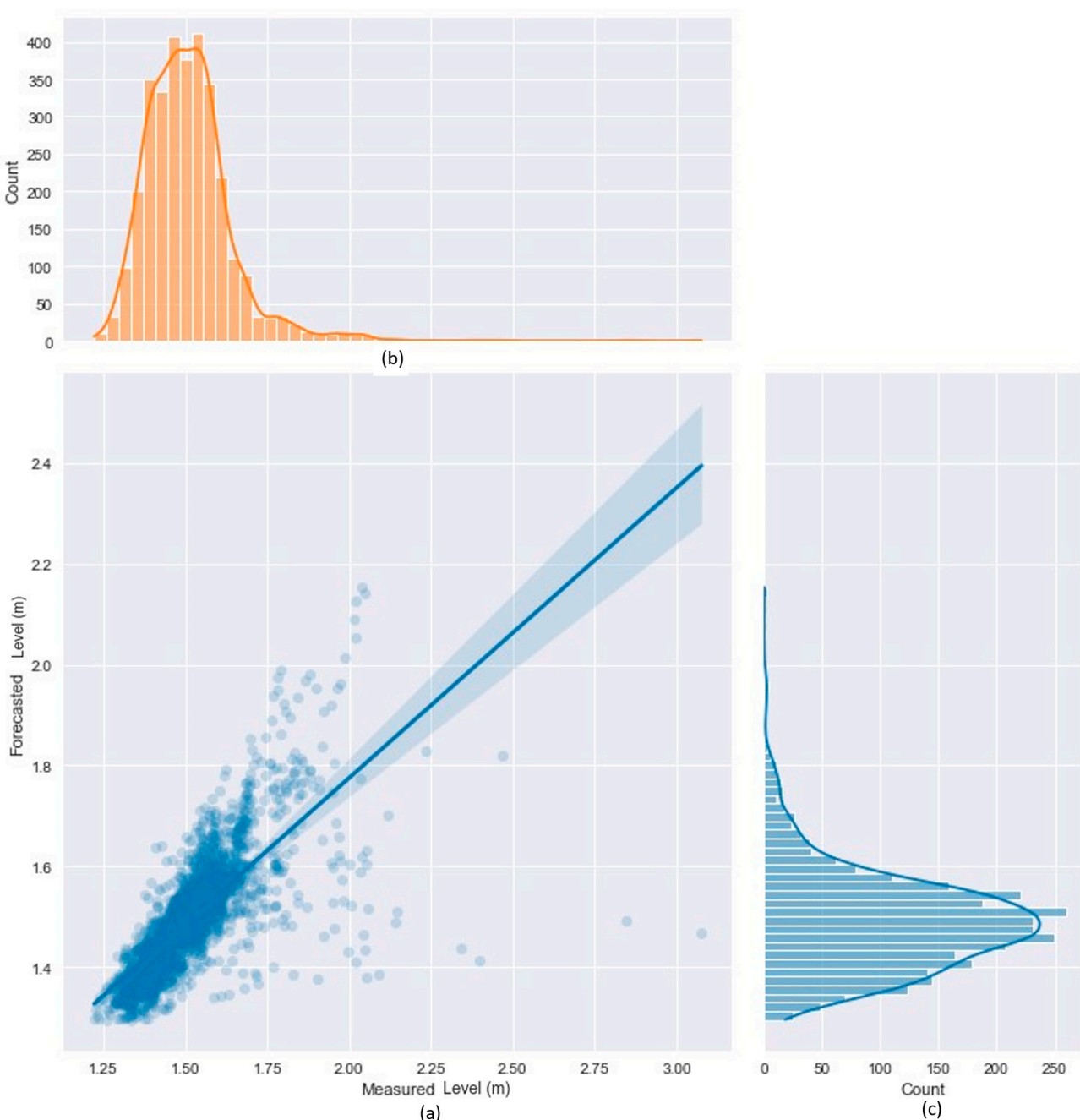

**Figure 22.** Scatter plot (**a**) with a marginal distribution for the measured (**b**) and forecasted (**c**) level values using GNN-SAGE for 12 h ahead.

Figure 22 shows the scattered points around the regression line. In this case, the forecasted data were mostly within the 1.35 m to 1.80 m interval, resulting in a high variance for the model's prediction, leading to more dispersed points, and thus, again reducing $R^2$ to a value of 40.12%.

### 3.8. Results for a 24 h Forecast Horizon

Considering a forecasting horizon of 24 h, the results found for the effects of different time lags on the models' performances are shown in Figure 23.

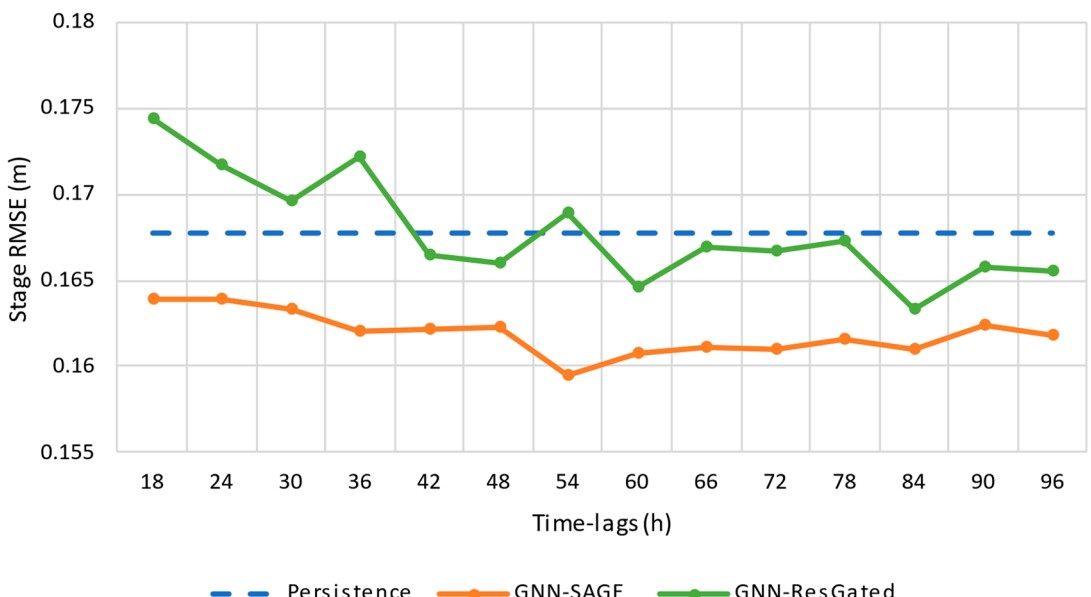

**Figure 23.** Influence of the different time lags for the GNN-SAGE and GNN-ResGated models for a 24 h forecasting horizon compared with the persistence model.

For one-day-ahead forecasting, increasing the time lag improved both the GNN-SAGE and GNN-ResGated predictions: the former, again, reached the best overall performance, converging its results for a 60 h time lag, resulting in an RMSE equal to 0.16079 m; meanwhile, the latter achieved its best result for an 84 h time lag and an RMSE equal to 0.16331 m. Interestingly, GNN-ResGated could not outperform the persistence model until a 42 h time lag, requiring more past information to improve its performance. Figure 24 presents the results of different input variables used for each model.

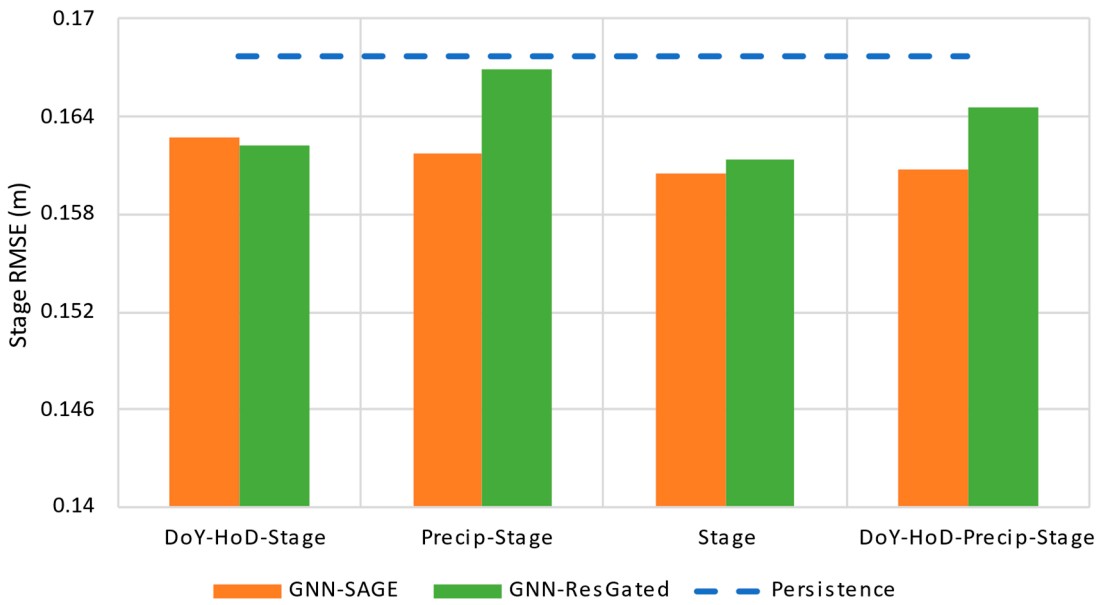

**Figure 24.** Influence of the different input variables for the GNN-SAGE and GNN-ResGated models for a 24 h forecasting horizon.

For the 24 h forecasting horizon, additional precipitation data added no significant information for the GNN-SAGE model. It reached similar results when compared with forecasting using just stage as an input variable, indicating that it may add noise rather than relevant information for long forecasting horizons. This may have been due to a combination of both the large time lag and forecasting horizon: due to their extended lengths, part of the data coming from upstream stations may not get captured by the predictive model, thus not improving its performance.

Due to the very similar results using just stage and all variables as inputs, there was no conclusive best configuration for this situation. However, to keep following the previous results from 3 h to 12 h forecasting horizons, the authors opted to use all independent variables as input, leading to an error of 0.160 m and an improvement of 4.13% over the persistence model and 2.28% over GNN-ResGated. Figure 25 shows the results for the accumulated precipitation from 1 h to 24 h.

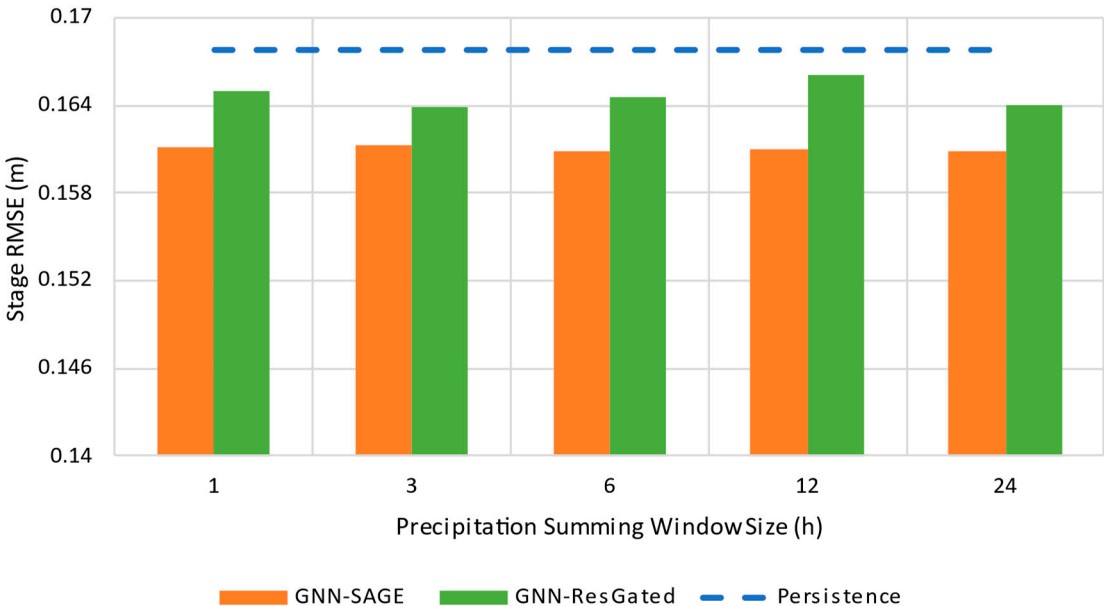

**Figure 25.** Influence of the different precipitation summing window sizes for a 24 h forecasting horizon.

Both GNN-SAGE and GNN-ResGated showed low sensitivity for different precipitation window sizes. The proposed GNN-SAGE model managed to achieve the best results for each one of the windows, with marginal differences between them. The minimum error of 0.16079 m was reached using a 6 h window. The GNN-SAGE performance was also evaluated based on how its forecasted results followed the real measured ones, as depicted in Figure 26.

Unsurprisingly, the 24 h ahead prediction led to the worst results thus far. From Figure 26, it is possible to see that the model completely missed the highest peak of more than 3.0 m. This indicated that the model could not correctly predict stage values too far into the future, with it not being able to provide precise results. The following Figure 27 illustrates the variance in the forecasted results around the regression line.

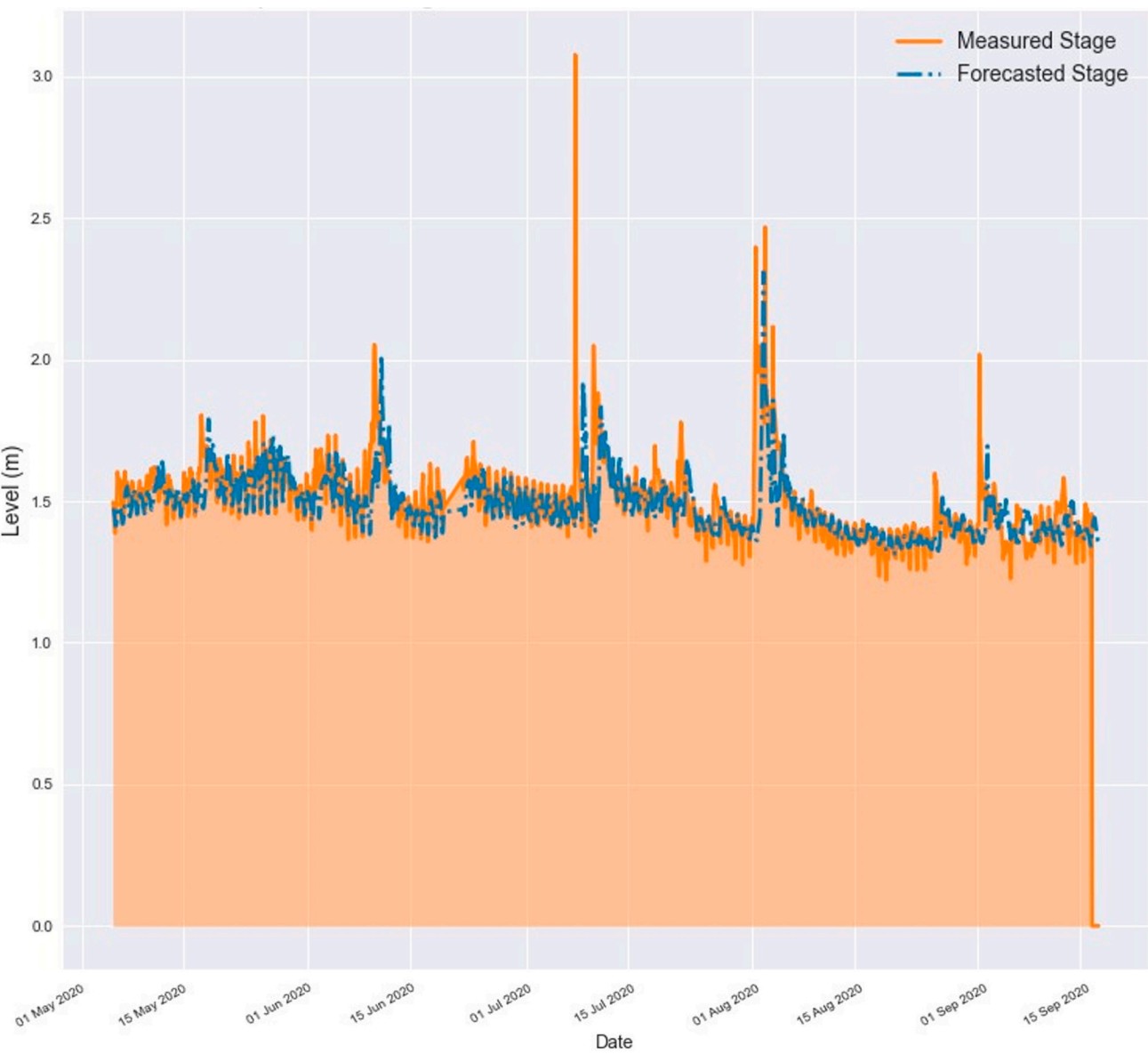

**Figure 26.** Comparison between the forecasted stage values using GNN-SAGE and the real measured stage values for Humber River for a 24 h forecasting horizon.

As expected, there was much more prominent scattering than the previously shown results. The high variance of the model for 24 h forecasting resulted in more dispersed forecasted points, indicating a less precise model for this situation. This was reflected in the model's $R^2$, which was lower than the previously assessed configurations, with a value of 22.65%.

### 3.9. Results of the SHAP Analysis

Figure 28a–e present the results of the SHAP analysis conducted for the proposed model. The SHAP analysis organized the variables in descending order of influence over the forecasted result. This meant that the closer to the top the variable was, the more influential it was. This approach also presented the correlation between input variables and the achieved result, as presented by the right sidebar in Figure 28a–e: the higher the feature value, the higher the correlation with the forecasted stage for the river. Besides this, positive values for the SHAP analysis indicated that the variable positively affected the model's forecasting, and vice versa.

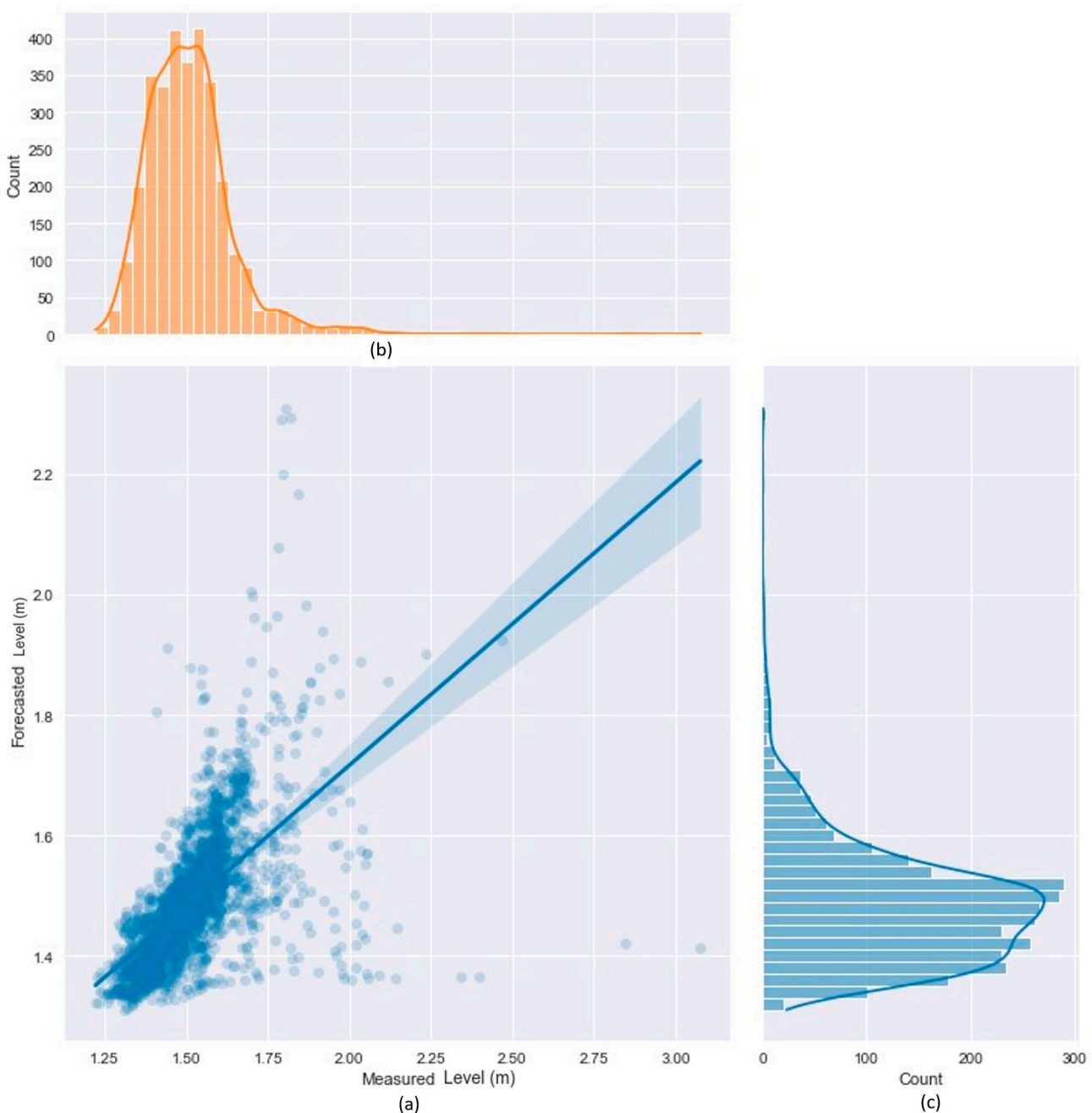

**Figure 27.** Scatter plot (**a**) with a marginal distribution for the measured (**b**) and forecasted (**c**) level values using GNN-SAGE for 24 h ahead.

From Figure 28, the most influential variable for the model for all forecasting horizons was related to the reference station itself, as indicated by the variable "Station 0_lag". This variable refers to the stage level of the reference station and shows a high correlation with the final output, negatively affecting the model's prediction. In Figure 28a, all the most influential variables were previous information from neighboring stations because only past-stage values were used for this configuration. However, as previously mentioned, from the 3 h forecast horizon onward, other input variables were added to the model. Figure 28b–e show that HoD and DoY played a significative role in the model's predictions, being amongst the most influential variables for this forecasting horizon. These variables could capture seasonality and diuturnal variations, adding important information for the model's forecasting due to the daily and yearly river stage variations.

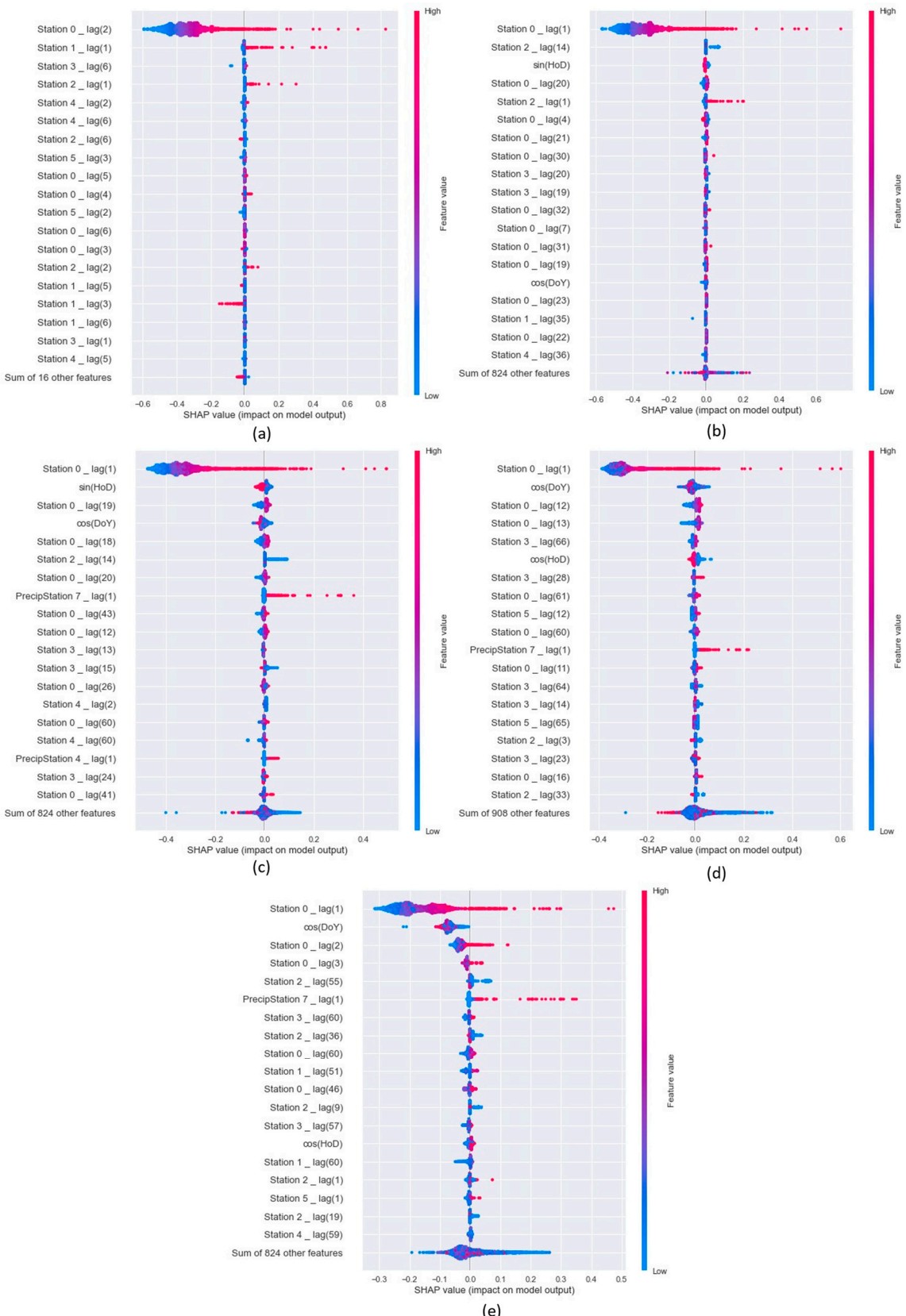

**Figure 28.** SHAP analysis results for (**a**) 1 h, (**b**) 3 h, (**c**) 6 h, (**d**) 12 h, and (**e**) 24 h horizons.

For the precipitation input data, its influence started to be relevant for the 6 h horizon onward, as shown in Figure 28c–e. Past information for precipitation coming from station 7,

e.g., "PrecipStation 7_lag", carried more spatiotemporal information than the other stations. Only for the 6 h forecasting horizon did another station provide influential information regarding precipitation, i.e., station 4.

## 4. Discussion

The proposed GNN-SAGE model managed to satisfactorily forecast flooding events of the Humber River up to 3 h in advance. After this time horizon, the results started to deteriorate, as shown in Figures 17, 22 and 27. However, this is an expected behavior in ML approaches that use time-series forecasting. Previous works on flooding forecasting showed that the error increased as the leading time increased, deteriorating future flood predictions and indicating that the input information may not provide enough spatiotemporal data for the model [24,80]. This may also lead to lagged forecasted results when compared with the expected values, as shown in Figures 16, 21 and 26 [81]. The performance deterioration could potentially be attributed to the dataset size, suggesting that more data might be required for increased precision in longer-range forecasts [24,79].

GNN-SAGE's performance was comparable with some of the state-of-the-art approaches found in the literature. Tables 1 and 2 summarize the performance metrics achieved by the proposed graph-based model and the results from previous works.

**Table 1.** Summary of performance metrics for forecasting using the GNN-SAGE model.

| Metric | 1 h Ahead | 3 h Ahead | 6 h Ahead | 12 h Ahead | 24 h Ahead |
|---|---|---|---|---|---|
| RMSE (m) | 0.02516 | 0.06736 | 0.09200 | 0.12215 | 0.16077 |
| MAE (m) | 0.01592 | 0.0345 | 0.04372 | 0.05514 | 0.07077 |
| MAPE | 1.04% | 2.18% | 2.65% | 3.20% | 3.78% |
| $R^2$ | 96.45% | 75.50% | 59.10% | 40.12% | 22.65% |

**Table 2.** Literature values for flooding prediction.

| Model | Metric Value | Author |
|---|---|---|
| Spatio-temporal attention LSTM (STA-LSTM) | Error rate<br>3.96% for 6 h forecasting horizon<br>3.98% for 12 h forecasting horizon<br>6.31% for 24 h forecasting horizon | Zhang et al.<br>[19] |
| Quantitative precipitation forecast (QPF) | RMSE<br>0.09 m for 1 h forecasting horizon | Wu et al.<br>[28] |
| Support vector machine (SVM) | RMSE (MAE)<br>0.072 m (0.036 m) for 3 h forecasting horizon<br>0.131 m (0.070 m) for 6 h forecasting horizon | Dazzi et al.<br>[80] |
| Support vector regression (SVR) | RMSE<br>0.07 m for 1 h forecasting horizon<br>0.25 m for 3 h forecasting horizon | Nguyen and Chen<br>[82] |
| Multiple additive regression trees (MART) | RMSE<br>0.14 m for 1 h forecasting horizon<br>0.29 m for 3 h forecasting horizon | Fu et al.<br>[83] |
| Hybrid wavelet and ANN (WANN) | RMSE ($R^2$)<br>0.03 m (98%) for 1 h forecasting horizon<br>0.038 m (97%) for 3 h forecasting horizon<br>0.12 m (60%) for 6 h forecasting horizon | Alexander et al.<br>[84] |

The hybrid approach STA-LSTM, as implemented by Zhang et al. [19], was used to forecast flooding of the Humber River. The attention-based LSTM model showed good agreement between real and predicted values for forecasting horizons ranging from 6 h to 24 h. When comparing GNN-SAGE with their results, it is possible to see that the

graph-based model achieved better results for each of the studied horizons. The results indicate that the GNN-SAGE had better performance when forecasting flooding of the Humber River than the attention model.

In Wu et al. [28], a physics-based model was implemented to forecast flooding for 1 h ahead in Taiwan. The authors managed to improve their results by 80% when compared with direct simulated values of the river using real-time data correction. However, when comparing GNN-SAGE with the best QPF result achieved, the graph-based model showed superior performance, achieving an RMSE that was 3.5 times smaller than the one presented in the literature.

The GNN-SAGE performance was also superior when analyzed against SVM and SVR models. In Dazzi et al. [80], the authors used an SVM to predict the river stage in Italy up to 9 h into the future. Their best results were found for the 3 h forecasting horizon, where the RMSE was 0.072 m and the MAE was 0.036. For a 3 h forecasting horizon, the errors were greater (as expected), achieving 0.131 m for the RMSE and 0.070 m for the MAE. When comparing these values with the ones presented in Table 1, GNN-SAGE surpassed the literature numbers for both assessed horizons, improving the forecasting for 3 h and 6 h ahead by 6% and 29%, respectively, in terms of RMSE. For the MAE, GNN-SAGE improved the forecasting by 4% for 3 h ahead and 38% for 6 h ahead. For both metrics, GNN-SAGE presented forecasted results more accurately for the river stage. When comparing with the work done by Nguyen and Chen [69], where the authors forecasted flooding events in Taiwan, the proposed graph-based model improved the RMSE metric by 64% and 73% for 1 h and 3 h, respectively.

The GNN-SAGE model also surpassed the results achieved by MART and hybrid WANN models. The MART model was used by Fu et al. [83] to predict flooding in China. Considering their downstream station and the best results from their study, which are presented in Table 2, GNN-SAGE presented an improvement of 82% for the 1 h forecasting horizon and 77% for the 3 h forecasting horizon, indicating great superiority over the MART model. Finally, in Alexander et al. [84], a hybrid WANN model was implemented to estimate flooding in India. Their model RMSE and $R^2$ were consistent between 1 h and 3 h ahead, with a minimal difference between them, as presented in Table 2. However, for the 6 h forecasting horizon, their RMSE and $R^2$ deteriorated greatly. Compared with the WANN model, GNN-SAGE improved the forecasting for 1 h ahead regarding RMSE by 16%. However, for this same forecasting horizon, the $R^2$ metric for the WANN model was superior, indicating that their model was better at explaining the data. For a 3 h forecasting horizon, GNN-SAGE was surpassed by the WANN model in terms of RMSE and $R^2$. However, our proposed model improved the forecasting for the 6 h horizon by 23%, while the $R^2$ was around the same value as the one in the literature.

The SHAP analysis showed that for most of the assessed horizons, past data from the reference and its neighboring stations were found to be the most relevant for the model forecasting. The river seasonality was captured using HoD and DoY, which started to exert relevant influence over GNN-SAGE from 3 h onward. Precipitation was also another relevant variable for the model starting from the 6 h horizon. Precipitation is an important parameter for hydrological event simulations [19,40,85], and accumulated precipitation data from the previous 72 h can give enough hydrological information for flooding forecasting [30]. In [86], SHAP analysis showed that forecasting streamflow relies on important precipitation data, besides streamflow inputs, often influencing the mode positively. In Ekmekcioğlu et al. [87] and Aydin and Iban [88], SHAP analysis showed that precipitation may affect the forecasted result differently depending on the ML model used, which is an expected behavior since different ML approaches process data differently, resulting in different results for an identical task [33,48,49,89].

## 5. Conclusions

The present work presented a graph-based deep-learning forecasting model to estimate flooding. The proposed model was validated for the Humber River, Canada. Using

historical data from stations distributed along the river, comprising previous information on the stage level and precipitation ranging from 8 June 2018 to 17 September 2020, the GNN-SAGE model was trained and tested to forecast possible flooding events up to 24 h in advance.

The forecasted results were assessed using the metrics RMSE and the coefficient of determination $R^2$. The RMSE values for 1 h, 3 h, 6 h, 12 h, and 24 h forecasting horizons were, respectively, 0.025 m, 0.067 m, 0.920 m, 0.122 m, and 0.160 m. For the coefficient of determination, the found values were 96.45%, 75.50%, 59.10%, 40.12%, and 22.65%, respectively, meaning average improvements over the benchmark persistence and GNN-ResGated models of 18% and 9%, respectively. The model presented a great performance for forecasting horizons of 1 h and 3 h. However, results deteriorated after that, which was an expected behavior due to the increase in uncertainty for longer forecasting horizons.

When compared with similar studies found in the literature, GNN-SAGE surpassed or provided competitive results in terms of errors RMSE, MAE, and MAPE, and the coefficient of determination $R^2$, except when compared with the hybrid WANN model. The graph-based model surpassed the physic-based QPF model, the ML SVM and SVR models, and the ensemble MART model, offering an average improvement of 53% over the literature models.

The SHAP analysis showed that the most influential variable for the model result was the data coming from the reference station for all assessed horizons. However, seasonality regarding DoY and HoD increased their importance over the forecasted values when the forecast horizon was increased. Precipitation was also more influential in the forecasted result starting from a forecasting horizon of 6 h.

In future works, the performance of GNN-SAGE can be investigated further when applied to a larger river body. In this case, the impact of spatiotemporal information from neighboring stations can be assessed by examining a variety of station numbers. The developed model can also be applied to an early alarm flooding system. The results of the graph-based model indicate that it can provide adequate warning ahead of time, allowing residents and stakeholders to be better prepared for flooding events.

**Author Contributions:** Conceptualization, J.V.G.T. and B.G.; methodology, P.A.C.R., J.V.G.T. and B.G.; software, P.A.C.R.; validation, P.A.C.R., J.V.G.T. and B.G.; formal analysis, P.A.C.R.; investigation, P.A.C.R., J.V.G.T. and B.G.; resources, J.V.G.T. and B.G.; data curation, J.V.G.T. and B.G.; writing—original draft preparation, V.O.S., P.A.C.R. and J.S.; writing—review and editing, V.O.S., P.A.C.R., J.S., J.V.G.T. and B.G.; visualization, V.O.S. and P.A.C.R.; supervision, J.V.G.T. and B.G.; project administration, J.V.G.T. and B.G.; funding acquisition, B.G. and J.V.G.T. All authors have read and agreed to the published version of the manuscript.

**Funding:** This research was funded by the Natural Sciences and Engineering Research Council of Canada (NSERC) Alliance, grant No. 401643, in association with Lakes Environmental Software Inc., and by the Conselho Nacional de Desenvolvimento Científico e Tecnológico—Brasil (CNPq), grant no. 303585/2022-6.

**Data Availability Statement:** The used data were acquired from https://backup.trcagauging.ca/ (accessed on 30 March 2023). The used algorithm can be downloaded from https://drive.google.com/drive/folders/1dCCG8tNBpXNJ_4do5OeXzYI4VnGwYGSp?usp=sharing (accessed on 5 April 2023).

**Conflicts of Interest:** The authors declare no conflict of interest.

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
