# Peer review of "A New Graph-Based Deep Learning Model to Predict Flooding with Validation on a Case Study on the Humber River"

_water, doi:10.3390/w15101827_

Round 1

Reviewer 1 Report

1. The GNN-SAGE model detail is missing. This is your main tool but still no directions for the future researchers to learn and understand this model. Also references are missing for someone to understand it from.

2.The paper is too long. Essentially, you are repeating stuff for 1 hr, 3hr, 6hr, 12 hr, and 24 hr. As journal space is precious, you can figure out a way to present information in a more concise manner.

3. Too little information was given about the persistence model. Surely, a few more sentences could be added since not all would refer to publications [58-60] for clarification.

4. Provide future research directions that you would like to pursue but could not. This would help researchers following your footsteps.

5. Fig 3.6 and Fig 3.11 and so on are composite figures. Each should be numbered e.g. Fig 3.6(a), (b) and (c) and there should be caption for each. Further, labels on x and y axis are incomplete. They should be ‘Forecasted/Measured stage (m)’ . Similarly, the histograms label for x-axis is missing and for y-axis is ‘counts’. Surely, more precision and rigour is required.

Plz see attached file for some minor issues

Author Response

Dear reviewer,

Please see the attachment for the point-by-point answer for your comments.

Best regards,

The authors.

Reviewer 2 Report

The manuscript can be accepted after minor revision. The introduction part gives detail information about literature and objectives of the study.

Materials and methods were well explained.

Results were well explained with graphs.

Discussion is well suported with results.

Conclusion might be improved by suggesting what can be done to improve the model and suggestions for future work.

Author Response

(The authors gave the same response as above.)

Reviewer 3 Report

Thank you for your outstanding work developing a deep-learning model to predict flooding. In terms of environmental impact, flooding is a critical topic. Your impressive research could impact society by improving flood preparedness and reducing flood damage.

Getting to the article:

Line 41.... To dimmish... Do you mean "To diminish"?

Also, you indicated your approach is valid, which seems to be the case. My question is, how can you possibly take your approach a step further? I was searching your content to see any future directions, implementation, or governmental application, but it seems I missed it. Could you please enlighten us about this? If you do not have such a discussion in the article, I believe adding some information would be helpful.

Otherwise, your article is scientifically sound and well-written. Some English checks may be required.

Line 41.... To dimmish... Do you mean "To diminish"?

Your article is scientifically sound and well-written. It was generally fluent, and I had no difficulty understanding your article. Some English checks may be required. 

Author Response

(The authors gave the same response as above.)
